# TTT3R: 3D RECONSTRUCTION AS TEST-TIME TRAINING

**Xingyu Chen**[1,2]    **Yue Chen**[1,2]    **Yuliang Xiu**[2]    **Andreas Geiger**[3]    **Anpei Chen**[2]
[1]Zhejiang University    [2]Westlake University    [3]Uni of Tübingen, Tübingen AI Center

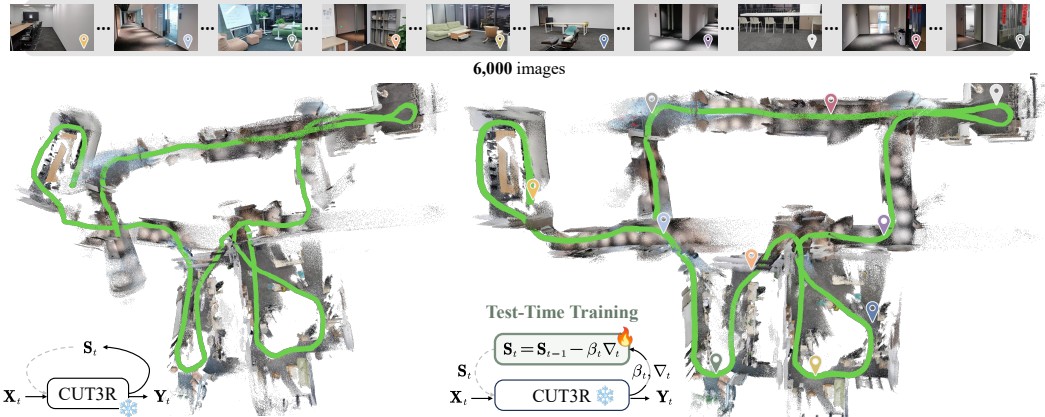

**Figure 1: Left:** CUT3R [89] encodes observations into a state (memory) $\mathbf{S}_{t-1}$, then interacts with new observation $\mathbf{X}_t$ and retrieves 3D information by reading out the output token $\mathbf{Y}_t$. However, it suffers from the forgetting problem and degrades significantly as the number of input views increases. **Right:** We treat the state $\mathbf{S}_t$ as a fast weight updated via gradient descent, where the learning rate $\beta_t$ and the gradient $\nabla_t$ are predicted by the frozen slow weights. These slow weights are learned from training datasets and act as a meta-learner, enabling the fast weight to serve as an associative memory. In addition, TTT3R makes online state updates by balancing the retention of historical information $\mathbf{S}_{t-1}$ with a confidence-aware learning rate $\beta_t$. This visualization also incorporates a state reset process, please see Appendix B for details.

## ABSTRACT

Modern Recurrent Neural Networks have become a competitive architecture for 3D reconstruction due to their linear-time complexity. However, their performance degrades significantly when applied beyond the training context length, revealing limited length generalization. In this work, we revisit the 3D reconstruction foundation models from a Test-Time Training perspective, framing their designs as an online learning problem. Building on this perspective, we leverage the alignment confidence between the memory state and incoming observations to derive a closed-form learning rate for memory updates, to balance between retaining historical information and adapting to new observations. This training-free intervention, termed TTT3R, substantially improves length generalization, achieving a 2× improvement in global pose estimation over baselines, while operating at 20 FPS with just 6 GB of GPU memory to process thousands of images. Code is available in `rover-xingyu.github.io/TTT3R`.

## 1 INTRODUCTION

3D reconstruction foundation models aim to predict camera poses and scene representations from a set of input RGB images. Building on the sequence modeling [76], recent advances [87, 91, 101] successfully map sequences of images into pixel-aligned pointmaps [11, 12]. Among these methods, the Transformer [82] has emerged as the dominant architecture, owing to its training efficiency and ability to capture long-range dependencies. However, a fundamental limitation lies in the quadratic growth of computational and memory costs with respect to sequence length. Despite various engineering optimizations, such as KV-cache compression [44] and flash attention [22], the softmax attention remains unchanged and continues to face limited scalability for long contexts [45].

Real-world applications often require handling an arbitrary number of images. As Figure 2 shows, recent feed-forward methods (e.g., VGGT [87], Point3R [95]) suffer from high memory consumption. Notably, only CUT3R [89] achieves constant memory usage with RNN-based design. However, as illustrated in Figure 1, CUT3R fails to generalize to long sequences due to training on most 64-frame sequences. Motivated by these observations, we ask ourselves if there are lessons from modern RNNs that can be used as design principles for 3D reconstruction.

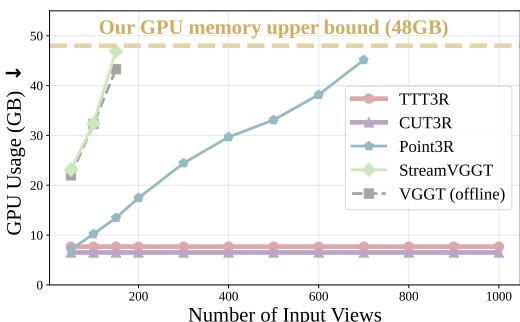

Figure 2: **GPU memory cost for inference.**

Recent advances in Recurrent Neural Networks (RNNs) demonstrate performance on par with Transformers on language tasks [34, 37]. Recurrent architectures compress the history context into a fix-length memory state, with each output depending solely on the current state and the incoming observation. This recurrent mechanism offers two benefits: efficient processing of long sequences with linear computational complexity, and the ability to scale to longer sequences by simply rolling out the state. Nevertheless, these benefits often come at the cost of substantial performance degradation, particularly when the sequence length exceeds the training context [62, 84].

This naturally raises two questions: (1) why do these models fail to provide robust length generalization? and (2) how can length generalization be achieved? To answer these questions, several studies [8, 62, 99] have investigated the length generalization of RNNs, identifying correlations with state overfitting [90], state forgetting [17, 36], and unexplored state distributions [62]. Solutions such as training on longer sequences and employing Truncated Backpropagation Through Time (TBTT) [62, 75, 94] have been proposed to improve length generalization. While these techniques have been incorporated into recent 3D reconstruction foundation models, such as CUT3R [89], they still struggle to generalize to sequences comprising hundreds of images.

In this work, we revisit the state update rule of recurrent 3D reconstruction models through the lens of Test-Time Training (TTT) [7, 73, 88], and systematically investigate the factors that hinder their ability to generalize across varying sequence lengths. Specifically, inspired by recent findings that recurrent models struggle with length generalization due to state overfitting [62], we reformulate state updating as a TTT-style online learning process [7, 45, 88]. In our framework, the historical information is compressed into a state online. We interpret the state as a fast weight [64, 65] learned at test time from the input in-context tokens, rather than from the training dataset. This perspective provides a principled understanding of state overfitting, suggesting that associative recall [9, 60] over long contexts, combined with gradient-based updates using adaptive learning rates to balance forgetting and learning [5, 6, 34, 45, 64, 74, 97], can substantially enhance length generalization.

Furthermore, we find that CUT3R [89] can be interpreted as a test-time training mechanism, whereas simply extending the sequence length during training leads to extremely low FLOPs utilization. Therefore, we propose a simple yet effective inference-time state update rule, termed **TTT3R**, derived as a closed-form state transition for online associative recall in CUT3R. This transition function explicitly defines the learning rate required to update the state at test time, thereby enabling length generalization. Our approach exploits internal confidence signals to selectively suppress low-quality state updates. This yields a stable, training-free gating mechanism that mitigates catastrophic forgetting [38] without requiring fine-tuning or additional parameters.

We evaluate TTT3R on standard 3D reconstruction benchmarks, which are typically configured with short-sequence inputs. In this setting, TTT3R performs competitively with state-of-the-art online reconstruction models [89, 95, 106] and demonstrates significant improvements with long-sequence inputs. More importantly, these gains in length generalization come at NO additional computational cost over the baseline, thanks to the proposed state update rule.

Overall, we introduce a new TTT-based framework to analyze the behavior of stateful 3D reconstruction models. Based on this, we propose a simple, empirical state update rule to enhance sequence length generalization for CUT3R.

## 2 RELATED WORK

**SfM and SLAM.** Structure-from-Motion (SfM) [2, 39, 57, 58, 66, 69, 70] and Simultaneous Localization and Mapping (SLAM) [23, 29, 43, 50, 52, 104] have long been the foundation for 3D structure reconstruction and camera pose estimation. These methods rely on associating 2D correspondences [4, 25, 47, 50, 63] or minimizing reprojected photometric errors [29, 30], followed by bundle adjustment (BA) [1, 12, 77, 79, 80, 86] for structure and motion refinement. Although highly effective when assembled into comprehensive systems [50, 66], these approaches often struggle in conditions of small camera parallax or ill-posed conditions (e.g., dynamic or textureless), leading to performance degradation. Recent work, such as MegaSaM [43] and VIPE [35], has demonstrated progress in adapting traditional SLAM paradigms to dynamic scenes by integrating semantic segmentation [35, 39], optical flows [35, 43, 104, 105], and geometric constraints [35, 39, 43, 48, 104]. Concurrently, methods like VGGT-SLAM [49] and VGGT-Long [24] seek improved robustness by integrating learned front-ends [51, 87, 91]. However, these methods require iterative optimization based on off-the-shelf estimation, where synchronization barriers often lead to cumulative errors and high computational overhead. This reliance hinders real-time online inference and learning scalability (e.g., the 'tabula rasa' blank slate limitation [89]). In this work, we investigate data-driven feed-forward models with generalizable priors to enable dense 3D reconstruction even from dynamic and textureless video sequences.

**Offline Reconstruction Foundation Model.** The pioneering feedforward 3D reconstruction method DUSt3R [91] introduced an end-to-end formulation that directly predicts two pixel-aligned pointmaps [11, 12, 68] from an image pair. By leveraging a Transformer-based architecture [28] and direct point supervision on large-scale 3D datasets, DUSt3R inherently accounts for image matching [3, 16] and pose estimation [27, 46], resulting in a reconstruction foundation model. Although some follow-up methods [16, 102] extend DUSt3R for robust dynamic scene reconstruction, they inherit the limitation of DUSt3R, requiring costly global alignment when the number of input views exceeds two. To address this issue, Fast3R [96] and VGGT [87] propose to use a large feedforward transformer with global attention that handles multiview inputs and predicts per-view pointmaps simultaneously, without the need for post-processing, leading to state-of-the-art 3D point and camera pose reconstruction. However, relying on the full attention [82] causes a quadratic increase in computational and memory cost, and results in an offline process that requires re-running inference over all images whenever a new frame arrives. Instead, we aim for compute-efficient, on-the-fly streaming inference that supports long-sequence, real-time interactive, and compute-efficient applications.

**Online Reconstruction Foundation Model.** To improve the reconstruction efficiency, several works introduce memory mechanisms to maintain information from past frames, enabling incremental reasoning and add pointmaps to a canonical 3D space. StreamVGGT [106] concurrently caches historical keys and values as memory in a causal transformer framework, allowing incremental processing. However, similar to full attention in VGGT [87], the computation and GPU memory usage in StreamVGGT grow redundantly. A promising alternative is the use of recurrent neural network architectures [14, 19, 85], such as CUT3R [89], which maintain a constant-sized memory state, while incrementally integrating new observations by simultaneously updating the state with the newly added view and retrieving historical information from the state. Although the recurrent formulation effectively reduces computational complexity and keeps inference memory usage consistently low, the memory-based methods suffer from the forgetting problem from earlier frames, leading to significant performance degradation as the number of input views increases. To mitigate the forgetting issue, Point3R [95] proposes an explicit point-based memory, where the history tokens are anchored in the reconstructed 3D point positions. While the explicit memory cache mitigates the forgetting, it causes memory cost that grows linearly as the number of views increases because the reconstructed points accumulate. In this work, we take an opposite path, exploring a closed-form state update rule that enhances the length generalization of implicit state memory to reasoning over thousands of views, while keeping memory and computation costs consistently low and unchanged as input view growth.

**Modern RNN.** Recent developments in RNN layers, serve as more efficient alternatives to quadratic complex full-attention layers [82], have demonstrated competitive performance in language modeling tasks. One line of research originates from a recurrent variant of attention [37, 64], known as linear attention [37], which uses the standard inner product between the query and key rather than the exponential softmax, allowing the output to be recurrently computed in linear time. However, linear

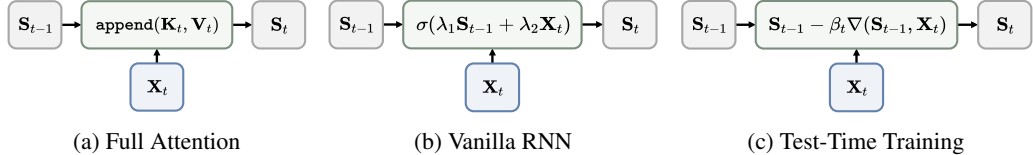

Figure 3: **Sequence Modeling Layers.** Full attention appends states, which incurs a quadratic cost. In contrast, vanilla RNNs use a fixed-size state with linear complexity, but they suffer from the forgetting problem. Our approach adopts Test-Time Training (TTT), treating the state as fast weights learned during test time via gradient descent with adaptive learning rates, which improves length generalization.

attention equally compresses all key value pairs into its finite-sized state, resulting in performance degradation as the sequence length increases. To address this limitation, various works [5, 54, 74, 97], such as Mamba [21, 34], have proposed adding forgetting gates, in which previous values are attenuated by a factor before the new memory is stored, to prevent the state from diverging over time. Recently, many of these models have been cast into a framework of test-time training/regression [7, 73, 88], which views the recurrent update of state as online learning [45] from context [26, 83], balancing between retaining historical memory and adapting to new information, as shown in Figure 3. The iterate states are also known as fast weights [64, 65], as they change in-context with each timestep, rapidly adapting to the input tokens. In contrast to the slow weights in neural networks—which act as meta-learners [31] and are only adjusted during training—fast weights are learned to function as associative memory [9, 60]. Recent examples of such layers include DeltaNet [64, 98], TTT [73, 103], and Titans [6], each of these layers was derived from a specific choice of retention and adaptation. The idea of learning at test time [10, 72] has also been explored in 3D reconstruction, where methods such as CVD [48] and Test3R [100] fine-tune a pre-trained model on the test sequence to minimize a self-supervised geometric consistency loss, thereby adapting the model to that particular scene. Inspired by their success, we propose to introduce a general test-time training framework that enables 3D reconstruction models to achieve both view scalability and memory retention.

## 3 METHOD

Our method processes a continuous stream of images received online. For each incoming image $\mathbf{I}_t \in \mathbb{R}^{W \times H \times 3}$, we aim to estimate, in real time and on the fly: the camera pose $\mathbf{T}_t \in \mathbb{R}^{3 \times 4}$, the camera intrinsic $\mathbf{C}_t \in \mathbb{R}^{3 \times 3}$, and the canonical point cloud $\mathbf{P}_t \in \mathbb{R}^{W \times H \times 3}$. We begin in Section 3.1 by introducing sequence modeling to understand and compare different prominent classes of methods that address the pointmap regression problem. Section 3.2 then reformulates recent incremental 3D reconstruction methods from a Test-Time Training perspective. Finally, in Section 3.3, we propose TTT3R, showing how the cross-attention between memory and observation can be leveraged as a confidence-guided state update rule for online associative recall.

### 3.1 SEQUENCE MODELING FOR POINTMAP REGRESSION

To transform a sequence of images to pixel-aligned pointmaps that lie in a unified global coordinate space, a generic formulation can be written as:

$$
\begin{aligned}
\mathbf{X}_t &= \texttt{Tokenize}(\mathbf{I}_t) \\
\mathbf{S}_t &= \texttt{Update}(\mathbf{S}_{t-1}, \mathbf{X}_t) \\
\mathbf{Y}_t &= \texttt{Read}(\mathbf{S}_t, \mathbf{X}_t) \\
\mathbf{P}_t &= \texttt{De-tokenize}(\mathbf{Y}_t)
\end{aligned}
\tag{1}
$$

where the input image $\mathbf{I}_t$ is patchified into a set of image tokens $\mathbf{X}_t \in \mathbb{R}^{(h \times w) \times c}$ through an image tokenizer [28], such as DINO [15, 53] and CroCo [92, 93]. The image tokens $\mathbf{X}_t$ update the previous state $\mathbf{S}_{t-1}$ into the current state $\mathbf{S}_t$ using new information. The model then retrieves information stored in the updated state $\mathbf{S}_t$ by reading out the output token $\mathbf{Y}_t \in \mathbb{R}^{(h \times w) \times c}$. Following the readout operation, the corresponding pixel-aligned 3D pointmaps are extracted via dense prediction de-tokenizers, such as linear layers with pixel shuffle [67] and a DPT head [61]. The camera pose $\mathbf{T}_t$ and the camera intrinsic $\mathbf{C}_t$, can either be solved from pixel-aligned 3D pointmaps using the

PnP [41] and Weiszfeld [56] algorithms, or regressed from image tokens $\mathbf{X}_t$ through the MLP or trunk attention layers [86, 87].

This sequence formulation offers a unified perspective for interpreting pointmap-oriented 3D reconstruction foundation models, where the update and read operations serves as the core distinction among different methods. They fall into two categories: full attention-based and RNN-based, each introducing specialized designs to the update rules of sequence modeling layers.

For full attention–based methods, such as Fast3R [96] and VGGT [87], all frames interact through global all-to-all self-attention, which can be interpreted as progressive state concatenation with growing state length:

$$
\begin{aligned}
\texttt{Update}(\mathbf{S}_{t-1}, \mathbf{X}_t) &= \mathbf{S}_{t-1}.\texttt{append}(\mathbf{K}_{\mathbf{X}_t}, \mathbf{V}_{\mathbf{X}_t}) \\
\texttt{Read}(\mathbf{S}_t, \mathbf{X}_t) &= \mathbf{X}_t + \texttt{softmax}(\mathbf{Q}_{\mathbf{X}_t}\mathbf{K}_{\mathbf{S}_t}^\top)\mathbf{V}_{\mathbf{S}_t}
\end{aligned}
\tag{2}
$$

where state $\mathbf{S}_{t-1} = [(\mathbf{K}_{\mathbf{X}_1}, \mathbf{V}_{\mathbf{X}_1}), \dots, (\mathbf{K}_{\mathbf{X}_{t-1}}, \mathbf{V}_{\mathbf{X}_{t-1}})]$ is a list of key-value pairs. Each key-value pair $(\mathbf{K}_{\mathbf{X}_t}, \mathbf{V}_{\mathbf{X}_t})$ and query $\mathbf{Q}_{\mathbf{X}_t}$ are transformed from the input token $\mathbf{X}_t$ via linear layers, and $\mathbf{K}_{\mathbf{S}_t} = \texttt{concat}[\mathbf{K}_{\mathbf{X}_1}, \dots, \mathbf{K}_{\mathbf{X}_t}]$, $\mathbf{V}_{\mathbf{S}_t} = \texttt{concat}[\mathbf{V}_{\mathbf{X}_1}, \dots, \mathbf{V}_{\mathbf{X}_t}]$. This modeling requires $\mathcal{O}(t^2)$ computing complexity, since all output tokens $\mathbf{Y}_1, \dots, \mathbf{Y}_t$ must be updated upon receiving $\mathbf{X}_t$. To efficiently process streaming input, StreamVGGT [106] uses a causal attention architecture to model the causal nature of streaming data, which restricts each frame to attend only to itself and preceding tokens, allowing only $\mathbf{Y}_t$ to be updated given $\mathbf{X}_t$. Causal attention enables incremental processing and reduces the computational cost to $\mathcal{O}(t)$. However, it shares a similar limitation with full attention: the state is represented as a key–value list that grows redundantly at $\mathcal{O}(t)$, leading to increasing memory consumption as the number of views increases.

For RNN-based methods [14, 19, 85, 89, 95], each incoming frame interacts with the state via one-to-one cross-attention, allowing for fixed-length state:

$$
\texttt{Update}(\mathbf{S}_{t-1}, \mathbf{X}_t) = \mathbf{S}_{t-1} + \texttt{softmax}(\mathbf{Q}_{\mathbf{S}_{t-1}}\mathbf{K}_{\mathbf{X}_t}^\top)\mathbf{V}_{\mathbf{X}_t}
\tag{3}
$$

where the state $\mathbf{S}_{t-1} \in \mathbb{R}^{n \times c}$, consisting of $n$ tokens with channel dimension $c$, encodes the scene with a constant length. $\mathbf{Q}_{\mathbf{S}_{t-1}}$ denotes the query projection obtained by applying a linear transformation to the state $\mathbf{S}_{t-1}$. Although this recurrent formulation effectively reduces the computational complexity to $\mathcal{O}(1)$ and inference memory usage constant at $\mathcal{O}(1)$, it suffers from forgetting and exhibits significant performance degradation as the number of input views increases.

## 3.2 REVISIT RNN-BASED RECONSTRUCTION THROUGH TTT

Test-Time Training (TTT) [73] introduces fast weights [65] as rapidly adaptable states that are updated during both training and inference to dynamically capture context. In contrast, slow weights (i.e., model parameters) remain frozen during inference. Formally, TTT represents the state as a fixed-length fast weight $\mathbf{S}_{t-1} \in \mathbb{R}^{n \times c}$ and updates it via gradient descent:

$$
\texttt{Update}(\mathbf{S}_{t-1}, \mathbf{X}_t) = \mathbf{S}_{t-1} - \beta_t \nabla(\mathbf{S}_{t-1}, \mathbf{X}_t)
\tag{4}
$$

where $\nabla(\mathbf{S}_{t-1}, \mathbf{X}_t)$ is a learned gradient function of the previous state $\mathbf{S}_{t-1}$ and the current observation $\mathbf{X}_t$, aiming to encourage the network to associate the current observation with the state, and $\beta_t$ is the learning rate. Intuitively, this online learning process encodes the KV-cache from the current observation into a fixed length of memory (i.e., state) as accurately as possible [88].

For example, linear TTT (or DeltaNet [64, 98]) minimizes the reconstruction error $\|(\mathbf{S}_{t-1}\mathbf{K}_{\mathbf{X}_t} - \mathbf{V}_{\mathbf{X}_t})\|^2$ by optimizing the state $\mathbf{S}$ to accurately reconstruct the observation value $\mathbf{V}_{\mathbf{X}_t}$. This objective yields an analytical gradient:

$$
\texttt{Update}(\mathbf{S}_{t-1}, \mathbf{X}_t) = \mathbf{S}_{t-1} - \beta \overset{\displaystyle (\mathbf{S}_{t-1}\mathbf{K}_{\mathbf{X}_t} - \mathbf{V}_{\mathbf{X}_t})\mathbf{K}_{\mathbf{X}_t}^\top}{\underset{\big\downarrow}{\nabla(\mathbf{S}_{t-1}, \mathbf{X}_t)}}
\tag{5}
$$

Here, the key $\mathbf{K}_{\mathbf{X}_t}$ serves as an index into previous state $\mathbf{S}_{t-1}$, identifying the entries to be updated with the value $\mathbf{V}_{\mathbf{X}_t}$. Conceptually, this mechanism treats the state as a dynamic associative memory, the key specifies *where to write*, the value specifies *what to write*, and the learning rate acts as a gating mechanism that controls the memory plasticity by *weighting* the intensity of the state update.

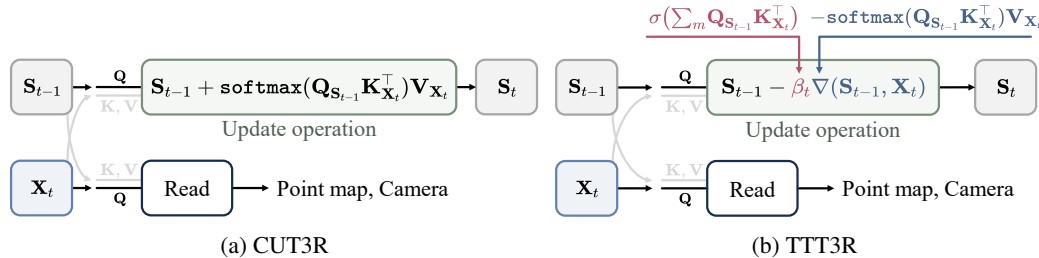

(a) CUT3R

(b) TTT3R

Figure 4: **TTT3R Illustration.** We present a training-free solution for scalable online 3D reconstruction that mitigates forgetting issue in CUT3R. (a) Vanilla CUT3R [89] pipeline. (b) Our reformulation from a test-time training perspective introduces a confidence-guided state update, where alignment confidence between memory and observations serves as per-token learning rates. See Eq. 8 for more details.

Next, we analyze the learning rate term $\beta$, which serves as the most critical hyperparameter and has been extensively studied in recent advances. It is typically represented as: 1) a learnable scalar parameter $\beta \in \mathbb{R}^1$ in RetNet [74]; 2) an input-dependent scalar function $\beta_t = \sigma\left(\ell_\beta\left(\mathbf{X}_t\right)\right) \in \mathbb{R}^1$ in DeltaNet [64, 98], TTT [73], and Mamba-2 [21]; and 3) a per-token function $\beta_t = \sigma\left(\ell_\beta\left(\mathbf{X}_t\right)\right) \in \mathbb{R}^{n \times 1}$ in Gated Linear Attention [97], which enables token-wise adaptive learning rates across all $n$ state tokens. Up to this point, we reformulate the Eq. 3 using the above TTT formulation:

$$\mathbf{S}_{t-1} + \texttt{softmax}(\mathbf{Q}_{\mathbf{S}_{t-1}}\mathbf{K}_{\mathbf{X}_t}^\top)\mathbf{V}_{\mathbf{X}_t} = \mathbf{S}_{t-1} - \beta_t \nabla\left(\mathbf{S}_{t-1}, \mathbf{X}_t\right) \tag{6}$$

The gradient is defined as a linear combination of the observation values $\mathbf{V}_{\mathbf{X}_t} \in \mathbb{R}^{(h \times w) \times c}$, weighted by the softmax alignment scores $\texttt{softmax}(\mathbf{Q}_{\mathbf{S}_{t-1}}\mathbf{K}_{\mathbf{X}_t}^\top) \in \mathbb{R}^{n \times (h \times w)}$ between the state query $\mathbf{Q}_{\mathbf{S}_{t-1}} \in \mathbb{R}^{n \times c}$ and the observation key $\mathbf{K}_{\mathbf{X}_t} \in \mathbb{R}^{(h \times w) \times c}$. Conceptually, the gradient function leverages cross-attention alignment between state query and observation key to determine *where to write*, assigning the corresponding observation value as *what to write* to each state token. This formulation has been demonstrated to be effective [91] for learning emergent functional 3D/4D alignment by implicitly matching cross-view context [16].

However, the softmax operation limits CUT3R's ability to balance retaining historical information with incorporating new inputs, as it forces the model to fully adapt to the latest observations. Specifically, because softmax weights are normalized to sum to 1.0 along the observation-token dimension, the model always prioritizes new information $\mathbf{X}_t$ over the historical state $\mathbf{S}_{t-1}$, leading to catastrophic forgetting. This forgetting also reflects the structural discrepancy relative to standard TTT: the lack of a flexible learning rate (effectively, a constant $\beta_t = 1.0$). Consequently, we are motivated to introduce a state update *weight* that serves as a gating mechanism to explicitly control memory plasticity.

### 3.3 CONFIDENCE-GUIDED STATE UPDATE RULE

Our core idea is to utilize alignment confidence between memory and observation to guide state updates. Figure 4 provides an overview of TTT3R. This confidence constitutes an adaptive per-token state update weight, serving as the per-token learning rate function in the TTT formulation [97].

Recall that the cross-attention (i.e., $\mathbf{Q}_{\mathbf{S}_{t-1}}\mathbf{K}_{\mathbf{X}_t}^\top$) aggregates information along the spatial dimension $m = \{1, \ldots, h\} \times \{1, \ldots, w\}$ of the image into $n$ state tokens. This process yields normalized attention weights for each state token, which are then used to compute a weighted sum over the value tokens $\mathbf{V}_{\mathbf{X}_t}$. To address the forgetting issue, we retain the original attention formulation but introduce a per-token learning rate $\beta_t \in \mathbb{R}^{n \times 1}$, derived from the alignment confidence between the state queries $\mathbf{Q}_{\mathbf{S}_{t-1}}$ and the observation keys $\mathbf{K}_{\mathbf{X}_t}$:

$$\beta_t = \sigma(\textstyle\sum_m \mathbf{Q}_{\mathbf{S}_{t-1}}\mathbf{K}_{\mathbf{X}_t}^\top) \tag{7}$$

To simplify notation, we define the summation $\sum_m$ to be normalized, thus representing a mean: $\sum_m \equiv \frac{1}{m}\sum_{i=1}^m$. This learning rate can act as a soft gate in gated attention, incorporating it into the

attention output allows for better long-context extrapolation [59]. Consequently, the full closed-form state update rule is given by:

$$\text{Update}(\mathbf{S}_{t-1}, \mathbf{X}_t) = \mathbf{S}_{t-1} - \underset{\color{red}\sigma(\sum_m \mathbf{Q}_{\mathbf{S}_{t-1}}\mathbf{K}_{\mathbf{X}_t}^\top)}{\beta_t} \overset{-\text{softmax}(\mathbf{Q}_{\mathbf{S}_{t-1}}\mathbf{K}_{\mathbf{X}_t}^\top)\mathbf{V}_{\mathbf{X}_t}}{\nabla(\mathbf{S}_{t-1}, \mathbf{X}_t)} \tag{8}$$

Note that, rather than ignoring quality variations and updating all state uniformly - which we find leads to suboptimal performance due to low-quality state updates (e.g., textureless regions, see Figure 5) - we leverage cross-attention statistics to estimate the alignment confidence of state updates and accordingly assign per-token learning rates $\beta_t$. That is, a higher alignment confidence in state updates generally indicates a stronger match between the state and observation with lower uncertainty, leading to a larger update step in our formulation. By aggregating token-level statistics, we suppress low-quality state updates to enhance performance. A similar

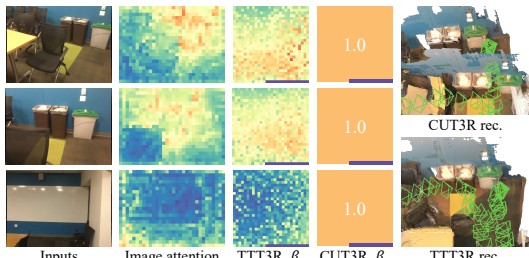

Figure 5: By incorporating image attention (i.e., $\mathbf{Q}_{\mathbf{S}_{t-1}}\mathbf{K}_{\mathbf{X}_t}^\top \in \mathbb{R}^{n \times (h \times w)}$) as per-token learning rates $\beta_t \in \mathbb{R}^{n \times 1}$, TTT3R mitigates catastrophic forgetting and facilitates online loop closure.

principle - leveraging internal confidence signals to selectively filter updates - has been explored in concurrent work [32] to improve test-time reasoning for large language models.

This formulation enables a training-free, plug-and-play intervention for CUT3R, which can be directly applied to downstream tasks without additional fine-tuning.

## 4 EXPERIMENTS

We evaluate our method on a variety of tasks, including camera pose estimation (Section 4.1), video depth estimation (Section 4.2), and 3D reconstruction (Section 4.3).

**Baselines.** We first compare TTT3R with the state-of-the-art online 3D reconstruction method CUT3R [89], which performs on-the-fly reconstruction with an RNN-based architecture over streaming images. We also evaluate against Point3R [95] and StreamVGGT [106], which extend CUT3R and VGGT [87], respectively, to longer sequences by fine-tuning with explicit pointmap memory or KV-cache–based state representations. In contrast, our approach introduces a general sequence modeling framework and an adaptive state learning rate, enabling a training-free solution. In the following experiments, we compare these methods in terms of reconstruction accuracy, GPU memory usage, and inference speed. See Appendix C.1 and Appendix C.2 of *Sup.Mat.* for more details.

### 4.1 CAMERA POSE ESTIMATION

Following prior works [89, 102], we evaluate camera pose accuracy on TUM dynamics [71] and ScanNet [20] datasets. We adopt the standard metric, Absolute Translation Error (ATE), computed after applying Sim(3) alignment [81] between the estimated and ground-truth camera trajectories.

The results of the long-sequence evaluation are shown in Figure 7. For reference, we also include VGGT, an offline method that can be considered as an upper bound for online methods, since its full attention mechanism preserves the entire history context without forgetting. We further evaluate inference efficiency in Figure 6 and Figure 2, reporting two metrics: frames per second (FPS) and peak GPU memory usage (GB). All models are evaluated on a single 48GB NVIDIA GPU, with the number of input views varied from 50 to 1000 and early termination if out-of-memory occurs.

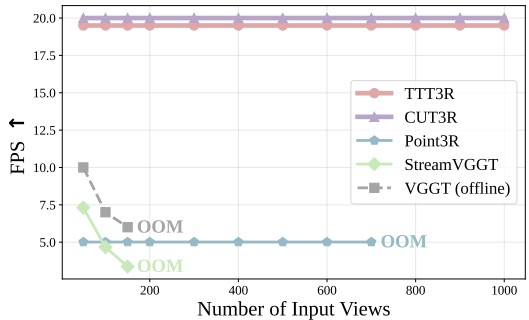

Figure 6: **Runtime** comparison on ScanNet [20]. *OOM* denotes the method out-of-memory beyond this point.

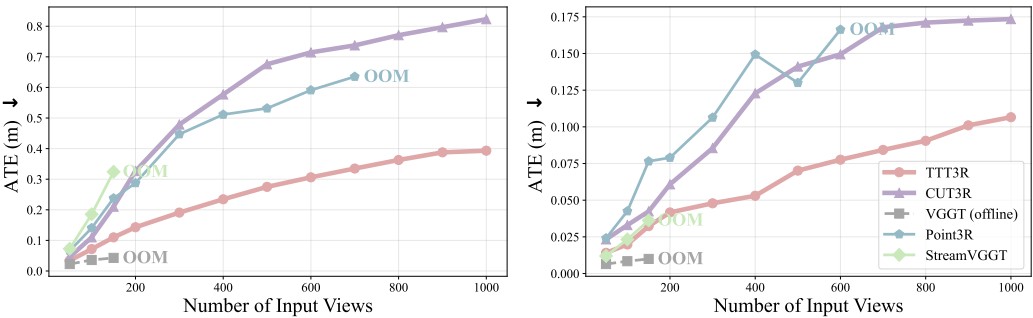

Figure 7: **Comparison of Camera Pose Estimation** on ScanNet [20] (left) and TUM-D [71] (right). *OOM* denotes out-of-memory. VGGT serves as an offline upper bound by preserving full history, yet full-attention methods (including StreamVGGT) suffer from high latency and memory exhaustion. Conversely, CUT3R is efficient but drifts on long sequences, while Point3R improves accuracy but hits OOM beyond 700 frames. Our method achieves a 2× accuracy improvement over CUT3R while retaining its real-time efficiency. See Appendix C.3 in *Sup.Mat.* for qualitative results.

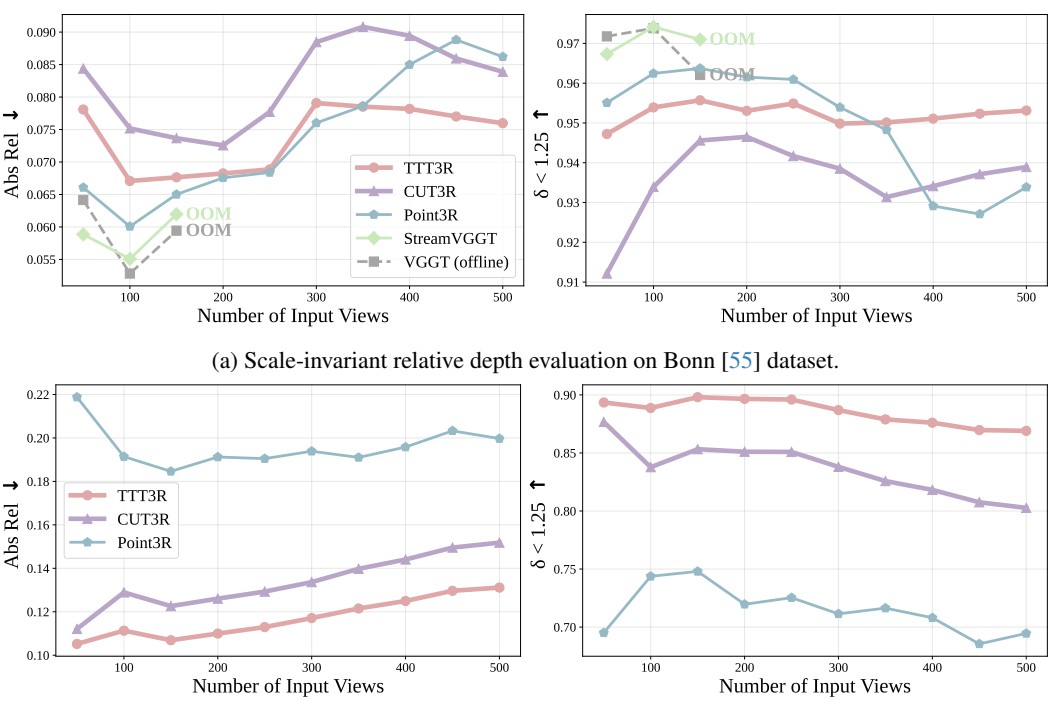

(a) Scale-invariant relative depth evaluation on Bonn [55] dataset.

(b) Metric depth evaluation on KITTI [33], excluding VGGT-based methods that don't support metric depth.

Figure 8: **Comparison of Video Depth Estimation.** *OOM* denotes out-of-memory. Full-attention methods (VGGT, StreamVGGT) serve as upper bounds for relative depth but hit OOM at > 150 frames. For metric depth, we compare the only online metric predictors: CUT3R, Point3R, and TTT3R. While Point3R achieves strong scale-invariant accuracy on short sequences (≤ 300 frames), it suffers from degradation on longer sequences and inaccurate metric prediction. In contrast, our approach consistently achieves the best overall performance without the need of fine-tuning.

As expected, VGGT and StreamVGGT, both based on full attention, are relatively slow and prone to memory exhaustion. CUT3R, in contrast, maintains consistently low GPU usage and real-time inference but struggles to retain information over long sequences, leading to inaccurate pose estimation. Point3R achieves improved accuracy over CUT3R by trading off GPU usage and runtime, but inference is slow and memory runs out beyond 700 frames. By reformulating CUT3R, our method achieves accurate pose estimation (with a 2× improvement) while preserving the same inference speed and memory efficiency as CUT3R.

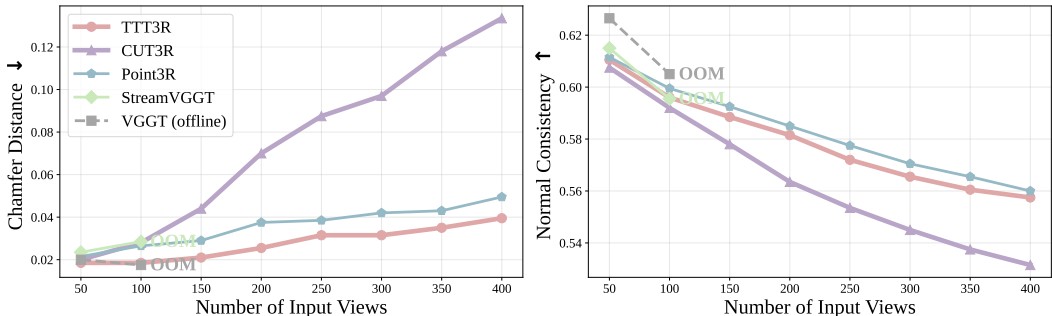

Figure 9: **Comparison of 3D Reconstruction** on 7-scene [68]. We evaluate geometric accuracy (Chamfer Distance ↓) and surface quality (Normal Consistency ↑) as the number of input views increases. Full-attention methods (VGGT, StreamVGGT) quickly exhaust memory (*OOM*). While CUT3R suffers from severe performance degradation due to forgetting, TTT3R maintains robust and consistent performance over long sequences, outperforming CUT3R significantly and achieving lower Chamfer Distance than Point3R. Figure 10 and Appendix C.5 in *Sup.Mat.* presents more qualitative results of 3D reconstruction.

Please refer to Appendix C.3 in *Sup.Mat.* for qualitative comparisons of camera trajectory estimation.

## 4.2 Video Depth Estimation

Following common practice [89, 102], we evaluate video depth estimation on the KITTI [33] and Bonn [55] datasets, which cover dynamic/static as well as indoor/outdoor scenes. We adopt standard metrics: absolute relative error (Abs Rel) and $\delta < 1.25$ (percentage of predicted depths within a 1.25-factor of true depth) . Video depth estimation measures both per-frame quality and inter-frame consistency, by aligning predicted depth maps to ground truth with a per-sequence scale, thereby evaluating relative depth accuracy. For methods that predict metric pointmaps (i.e., outputs in meters with absolute scale), we also report results without scale alignment, evaluating predictions directly in metric units to assess absolute-scale accuracy.

Figure 8 shows quantitative comparison against online baselines. VGGT and StreamVGGT run out of memory after about 150 frames due to their reliance on full attention. Nonetheless, they serve as an upper bound in per-sequence relative depth evaluation. For metric depth estimation, we report only CUT3R, Point3R, and TTT3R, as these are the only online methods predicting metric pointmaps.

Point3R achieves strong scale-invariant accuracy on short sequences ($\leq 300$ frames) due to its explicit pointmap memory, but suffers from forgetting and degraded metric-scale accuracy on longer sequences. In contrast, our approach consistently improves over baselines and achieves the best overall performance, without the need of fine-tuning. See Appendix C.4 of *Sup.Mat.* for more results.

## 4.3 3D Reconstruction

We follow previous work [85, 89] to evaluate 3D reconstruction on the 7-scene [68] dateset by measuring the distances between estimated pointmaps and ground-truth point clouds. As in prior work [18, 85, 89], we use chamfer distance and normal consistency as evaluation metrics. Chamfer distance is computed as the average of accuracy (nearest Euclidean distance from a reconstructed point to ground truth) and completeness (the reverse). Unlike prior approaches [85, 89], which sparsely sample 3–5 frames per scene, we evaluate performance on long image sequences to assess the memorization capability of different models.

Figure 9 shows that our method significantly outperforms other online approaches such as CUT3R [89] and StreamVGGT [106], and achieves results comparable to the top offline, full-attention method VGGT, while operating online in real-time with only 6GB GPU memory. This highlights the effectiveness of our method for 3D reconstruction. Figure 10 presents a qualitative comparison with CUT3R. TTT3R achieves more accurate reconstructions, whereas CUT3R suffers from catastrophic forgetting, leading to drifted camera poses, broken geometry, severe distortions, and ghosting artifacts. For more 3D reconstruction results, please refer to Appendix C.5 in *Sup.Mat.*

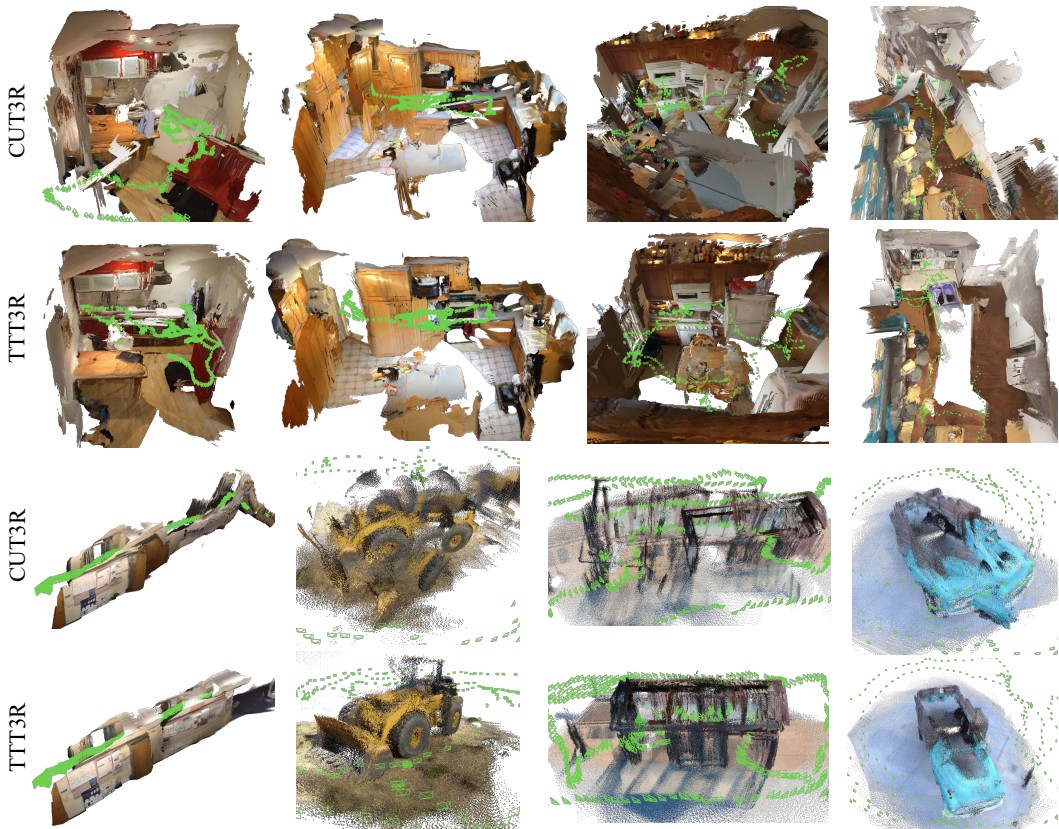

Figure 10: **Qualitative Results for 3D Reconstruction.** Compared to CUT3R, TTT3R improves sequence length generalization, mitigates forgetting, and enables online loop closure. Other baselines (e.g., VGGT, Point3R) are omitted due to OOM on long sequences. ▶ Check our website to see more video comparisons.

## 5 DISCUSSION

This paper presents TTT3R, providing a Test-Time Training perspective for recent 3D reconstruction foundation models, and proposes a simple yet efficient modification to CUT3R that enhances its length generalization. Our experiments demonstrate that TTT3R achieves robust long-sequence 3D reconstruction and outperforms state-of-the-art methods in most cases. The update is performed during the forward pass without model fine-tuning, making it a lightweight, plug-and-play solution.

**Limitations.** TTT3R mitigates but does not resolve state forgetting, and it has not yet matched strong offline methods (e.g., VGGT) in reconstruction accuracy, where full attention — despite being slower and more memory-demanding — preserves the entire history context. This behavior aligns with the *unexplored states hypothesis* [62], which posits that models trained on short contexts fail to generalize to longer sequences because their recurrence drives the state into out-of-distribution regions not encountered during training. To address this, we explore an optional TTT3R + State Reset variant (detailed in Appendix B in *Sup.Mat.*): by resetting the state to its initial value periodically, we effectively prevent state overfitting. These chunks are then aligned using global metric poses without additional optimization, offering a plug-and-play solution that retains the inference speed and memory efficiency of CUT3R.

**Future Work.** While TTT3R shows a clear boost of test-time regression for associative recall, its design space remains largely unexplored. Recent work [6, 7, 88, 103] highlights a vast opportunity to develop more effective, stable, and parallelizable recurrent architectures. We hope our findings will motivate future research to revisit the foundations of 3D reconstruction models and further improve the reconstruction accuracy and length generalization.

**Acknowledgments.** We thank the members of *Inception3D* and *Endless AI* Labs for their help. *Xingyu Chen* and *Yue Chen* are funded by the Westlake Education Foundation. *Xingyu Chen* is also supported by the Natural Science Foundation of Zhejiang province, China (No. QKWL25F0301). *Andreas Geiger* are supported by the ERC Starting Grant LEGO-3D (850533) and DFG EXC number 2064/1 - project number 390727645.

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

# Appendix

## Table of Contents

## A  MORE EXPERIMENTAL ANALYSIS

This section provides a detailed experimental analysis to validate the design choices of TTT3R. Specifically, we conduct three primary studies: first, we perform a **Comparison with Learnable Gating Mechanisms** to contrast TTT3R against existing test-time training baselines that employ different learnable gating mechanisms for modeling the state update learning rate; second, through the **TTT3R Finetuning Analysis**, we investigate whether applying our derived update rule and confidence-guided learning rate during the training process leads to further performance gains; and finally, we present a **Comparison with Non-TTT Baselines** to evaluate the core effectiveness of our Test-Time Training (TTT) reformulation by comparing it against a set of strong non-TTT methods.

### A.1  COMPARISON WITH STANDARD LEARNABLE GATING MECHANISMS

**Conceptual Relation to Common Gating Mechanisms.** To further analyze the relationship between our proposed confidence-guided learning rate and existing standard gating mechanisms, we compare how the learning rate $\beta$ is modeled in recent advances:

1. **ScalarLR**: In RetNet [74], the learning rate is represented as a single learnable scalar parameter, $\beta \in \mathbb{R}^1$. However, the constraint that all observations share the same learning rate inherently limits its representational flexibility.

2. **ConditionLR**: In models such as DeltaNet [64, 98], TTT [73], and Mamba-2 [21], the learning rate is modeled as an input-dependent scalar function: $\beta_t = \sigma\left(\ell_\beta\left(\mathbf{X}_t\right)\right) \in \mathbb{R}^1$. This provides the capability to condition the learning rate on the current observation $\mathbf{X}_t$.

3. **TokenLR**: Gated Linear Attention [97] further proposed a $\beta_t$ that is both input-conditioned and per-token: $\beta_t = \sigma\left(\ell_\beta\left(\mathbf{X}_t\right)\right) \in \mathbb{R}^{n \times 1}$. This enables adaptive assignment of individual learning rates for the $n$ state tokens, conditioned on the current observation.

Our core idea also models a conditioned per-token learning rate $\beta_t \in \mathbb{R}^{n \times 1}$, following the TokenLR paradigm. However, in contrast to previous gating mechanisms that rely on additional learnable parameters to model the learning rate (e.g., a scalar in ScalarLR, or a hyper-network $\ell_\beta(\cdot)$ in ConditionLR and TokenLR), we derive a training-free, closed-form learning rate. This is motivated by

the explainable cross-attention mechanism [16] and is derived directly from the alignment confidence between the state queries $\mathbf{Q}_{\mathbf{S}_{t-1}}$ and the observation keys $\mathbf{K}_{\mathbf{X}_t}$: $\beta_t = \sigma\left(\sum_m \mathbf{Q}_{\mathbf{S}_{t-1}} \mathbf{K}_{\mathbf{X}_t}^\top\right)$. This formulation achieves per-token adaptivity without introducing any additional parameters, training overhead, or computational cost.

**Practical Differences between Common Gating Mechanisms.** To further analyze the effectiveness of our proposed TTT-derived update rule and confidence-guided learning rate, we discuss the practical differences between TTT3R and simply incorporating a standard learnable gating module into the CUT3R state update. As detailed in Table 1 and Figure 11, we introduce ScalarLR, ConditionLR, and TokenLR into the state update of CUT3R. In this ablation study, we freeze the encoder, decoder, and output heads, and train only the newly added learnable gating module. The training is conducted using the same dataset as CUT3R, with

|  | Camera Pose | | Video Depth | |
|---|---|---|---|---|
| Method | ATE ↓ | RPE rot ↓ | Abs Rel ↓ | $\delta < 1.25$ ↑ |
| CUT3R | 0.173 | 0.494 | 0.152 | 80.2 |
| CUT3R + ScalarLR | 0.165 | 0.502 | 0.151 | 80.6 |
| CUT3R + ConditionLR | 0.166 | 0.509 | 0.149 | 81.0 |
| CUT3R + TokenLR | 0.154 | 0.497 | 0.148 | 81.5 |
| **TTT3R** | 0.106 | **0.431** | **0.131** | **86.9** |
| **TTT3R + Finetune** | **0.091** | 0.434 | 0.133 | 86.3 |

Table 1: **Evaluation** of camera pose estimation (1000 frames on TUM-dynamic [71]) and video depth estimation (500 frames on KITTI [33]). TTT3R achieves state-of-the-art performance across all compared learnable gating mechanisms. Finetuning TTT3R provides marginal benefits in pose estimation.

sequences ranging from 4 to 64 views. The results clearly demonstrate that CUT3R + TokenLR is the most effective among the learnable gating mechanisms. However, it still significantly underperforms TTT3R in both camera pose estimation and video depth estimation. In our observation, the performance of the learnable gating mechanisms is primarily limited by the training sequence length (i.e. 64 frames). We find that the longer the training sequence, the better the learnable gating mechanism performs. Unfortunately, training on sequences longer than 64 frames is prohibitively costly and becomes infeasible given our hardware constraints (48GB NVIDIA GPU). In contrast, our approach derives the learning rate directly from context without requiring expensive long-sequence training.

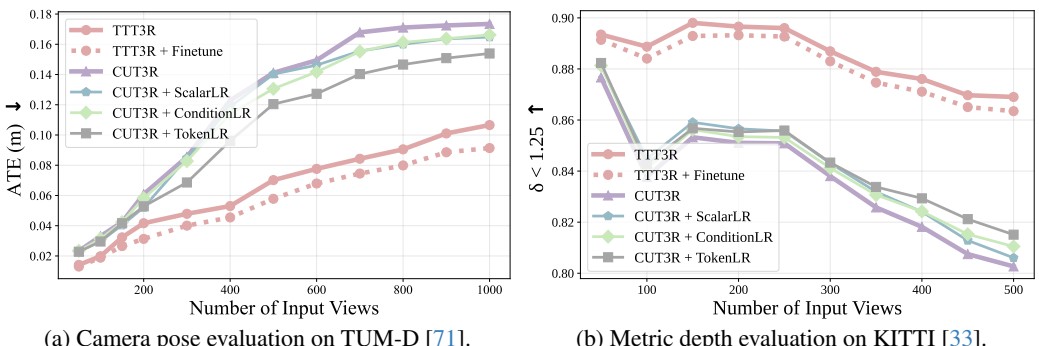

(a) Camera pose evaluation on TUM-D [71].  (b) Metric depth evaluation on KITTI [33].

Figure 11: **Quantitative Comparison of Camera Pose and Video Depth Estimation.** TTT3R achieves state-of-the-art performance compared to all baseline methods incorporating learnable gating mechanisms. Finetuning TTT3R yields marginal benefits in pose estimation.

## A.2 FINETUNING TTT3R

We then experiment with finetuning TTT3R by applying our derived update rule and confidence-guided learning rate during the training process. The training is conducted using the same dataset as CUT3R, with sequences ranging from 4 to 64 views.

As shown in Table 1 and Figure 11, finetuning leads to better performance in camera pose estimation, but concurrently degrades the video depth estimation results. One possible reason for this outcome, as illustrated in the CUT3R work, is that once the model is finetuned on 4-64 views, the model tends to prioritize global alignment over per-view prediction accuracy, which introduces forgetting and overall performance degradation of video depth estimation.

While incorporating the TTT3R update rule during training helps pose estimation, it does not provide significant benefits if the training sequences are short (e.g., 64 frames). As we can see from Figure 7, TTT3R shows only minimal performance gains compared to CUT3R on short sequences. We therefore hypothesize that scaling up the training sequence length could lead to better performance.

## A.3 TTT-DERIVED UPDATE RULE VS. NON-TTT BASELINES

To evaluate the effectiveness of our Test-Time Training (TTT) reformulation, we compare TTT3R against strong non-TTT baselines, including periodic state reset, Exponential Moving Average (EMA) shrinkage to the initial state, and burn-in mechanism with keyframes. We first establish the optimal hyperparameters for these baselines:

**1) CUT3R + Reset**. This mechanism performs a periodic state reset, where the state is reset to its initial value every $n$ frames. The resulting state chunks are then globally aligned using metric camera poses. We first ablate the choice of the reset period $n \in \{50, 100, 150\}$. As shown in Table 2, $n = 100$ yields the best results, which we adopt for subsequent experiments.

| Baselines | ATE ↓ | RPE trans ↓ | RPE rot ↓ |
|---|---|---|---|
| Reset 50 | 0.129 | 0.007 | 0.406 |
| Reset 100 | **0.126** | **0.007** | **0.403** |
| Reset 150 | 0.145 | 0.008 | 0.416 |

Table 2: **Ablation** on the periodicity of the State Reset baseline.

**2) CUT3R + EMA**. This method introduces shrinkage towards the initial state $S_0$ using an Exponential Moving Average (EMA) during inference. The state update is formulated as: $S_t = (1 - \alpha)S_{t-1} + \alpha S_0$, where $\alpha$ is the shrinkage rate. We ablate the choice of $\alpha \in \{0.0001, 0.001, 0.01\}$. As shown in Table 3, $\alpha = 0.001$ provides the optimal performance.

| Baselines | ATE ↓ | RPE trans ↓ | RPE rot ↓ |
|---|---|---|---|
| EMA 0.0001 | 0.169 | 0.007 | 0.574 |
| EMA 0.001 | **0.164** | **0.007** | **0.525** |
| EMA 0.01 | 0.191 | 0.009 | 0.673 |

Table 3: **Ablation** on the shrinkage rate of the EMA baseline.

**3) CUT3R + BurnIn**. The burn-in mechanism updates the state exclusively using keyframes, leaving the state unchanged for intermediate frames. We ablate the keyframe interval $n \in \{50, 100, 150\}$ frames. As shown in Table 4, an interval of $n = 100$ yields the best results, which we utilize for CUT3R + BurnIn.

| Baselines | ATE ↓ | RPE trans ↓ | RPE rot ↓ |
|---|---|---|---|
| BurnIn 50 | 0.150 | 0.0384 | 3.784 |
| BurnIn 100 | **0.144** | **0.0309** | **3.093** |
| BurnIn 150 | 0.156 | 0.0360 | 3.693 |

Table 4: **Ablation** on the keyframe interval for the Burn-In baseline.

As shown in Table 5 and Figure 12, TTT3R significantly outperforms all non-TTT baselines (Reset, EMA, and BurnIn) in both camera pose estimation and video depth estimation, thereby validating the effectiveness of our TTT-derived update rule. We note that these non-TTT mechanisms can also be integrated into TTT3R. Specifically, we observe that the Reset mechanism is highly effective for preventing state overfitting. Thus, we integrate Reset into TTT3R with the same optimal hyperparameter ($n = 100$), leading to the variant TTT3R + Reset. This combination further boosts TTT3R to achieve better performance. We provide a detailed analysis of this phenomenon in Section B.

| Method | Camera Pose | | Video Depth | |
|---|---|---|---|---|
| | ATE ↓ | RPE rot ↓ | Abs Rel ↓ | δ <1.25 ↑ |
| CUT3R | 0.173 | 0.494 | 0.152 | 80.2 |
| CUT3R + Reset | 0.126 | 0.403 | 0.128 | 84.9 |
| CUT3R + EMA | 0.164 | 0.525 | 0.150 | 80.7 |
| CUT3R + BurnIn | 0.144 | 3.093 | 0.151 | 80.2 |
| **TTT3R** | 0.106 | 0.431 | 0.131 | 86.9 |
| **TTT3R + Reset** | **0.093** | **0.375** | **0.115** | **88.5** |

Table 5: **Evaluation** of camera pose estimation (1000 frames on TUM-dynamic [71]) and video depth estimation (500 frames on KITTI [33]). TTT3R achieves better performance across all compared non-TTT baselines. Furthermore, the integrated variant, TTT3R + Reset, achieves the best performance.

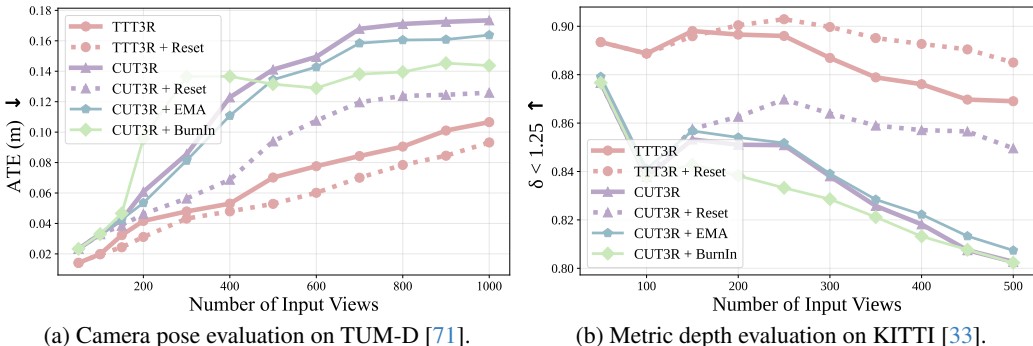

(a) Camera pose evaluation on TUM-D [71].   (b) Metric depth evaluation on KITTI [33].

Figure 12: **Quantitative Comparison of Camera Pose and Video Depth Estimation.** TTT3R achieves state-of-the-art performance compared to all non-TTT baselines. Furthermore, the integrated variant, TTT3R + Reset, achieves superior performance in both camera pose and video depth estimation.

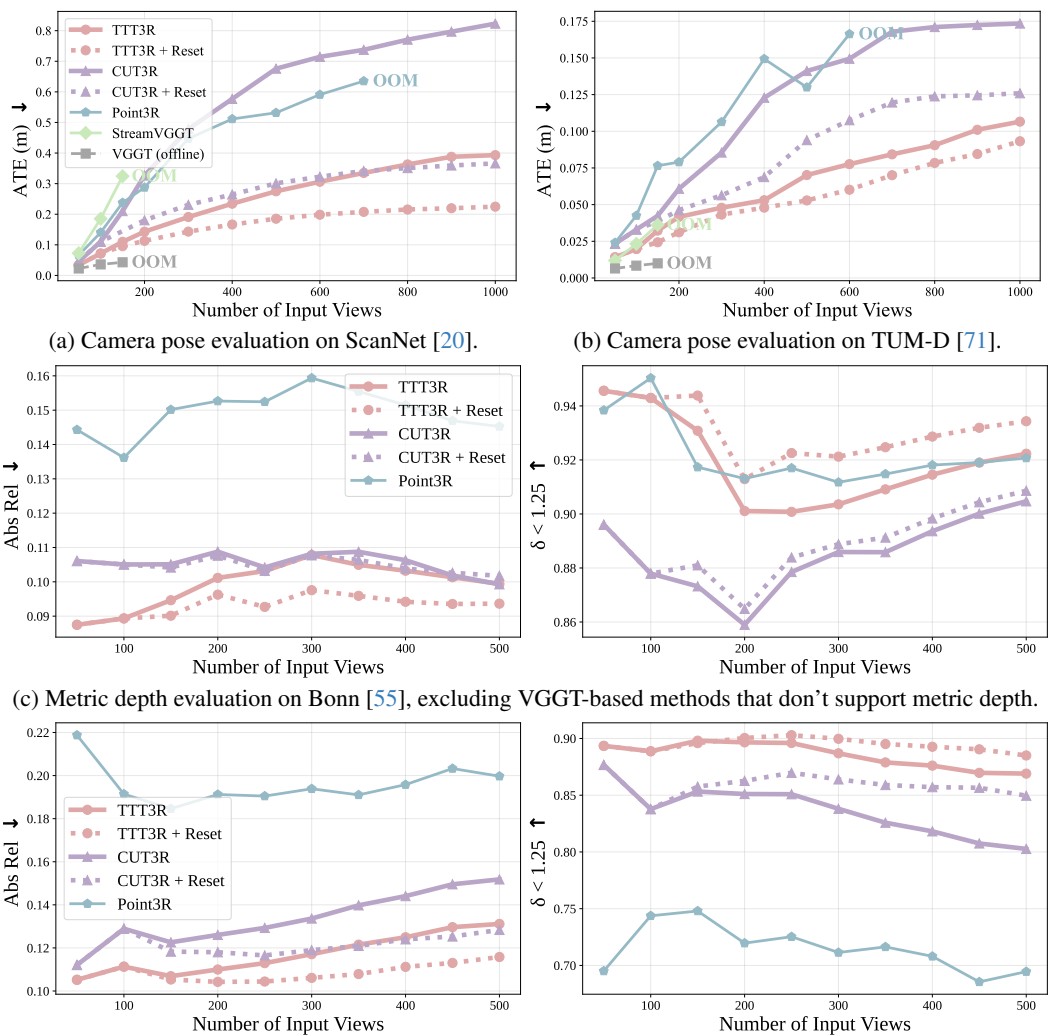

(a) Camera pose evaluation on ScanNet [20].  (b) Camera pose evaluation on TUM-D [71].

(c) Metric depth evaluation on Bonn [55], excluding VGGT-based methods that don't support metric depth.

(d) Metric depth evaluation on KITTI [33], excluding VGGT-based methods that don't support metric depth.

Figure 13: **Comparison of Camera Pose and Video Depth Estimation.** For the sake of simplicity, with the exception of Fig. 1, we **do not** use State Reset in any experiments or analyses reported in the main paper. For readers interested in the improvements achievable with State Reset, we provide comprehensive results here.

# B  STATE RESET

As discussed in the limitation subsection (Sec. 5), TTT3R only mitigates but does not fully resolve state forgetting, resulting in failure beyond 1000 frames (as shown in Fig. 14, middle). This observation is consistent with the unexplored states hypothesis [62], which posits that models fail to generalize to longer sequences when their recurrence, applied to extended sequences, produces state distributions not encountered during training—suggesting that these models overfit to states produced early in the sequence when trained on short contexts. Based on this hypothesis, we incorporate a State Reset mechanism: the state is reset to its initial value every 100 frames, thereby preventing state overfitting (as shown in Fig. 14, right). The resulting chunks are then aligned using the global metric camera poses without any optimization, making TTT3R + State Reset a plug-and-play solution that preserves CUT3R's inference speed and memory footprint.

Note that, for the sake of simplicity, with the exception of Fig. 1, we **do not** employ State Reset in any experiments or analyzes reported in the main paper. The State Reset is only used for visualization demonstrations that exceed 1000 frames. Specifically, we only apply the State Reset in Fig. 1 and Fig. 15—where we visualize the final result and also augment CUT3R with the State Reset—to allow for a fair and intuitive comparison. For readers interested in the improvements achievable with State Reset, we provide quantitative results in Figure 13.

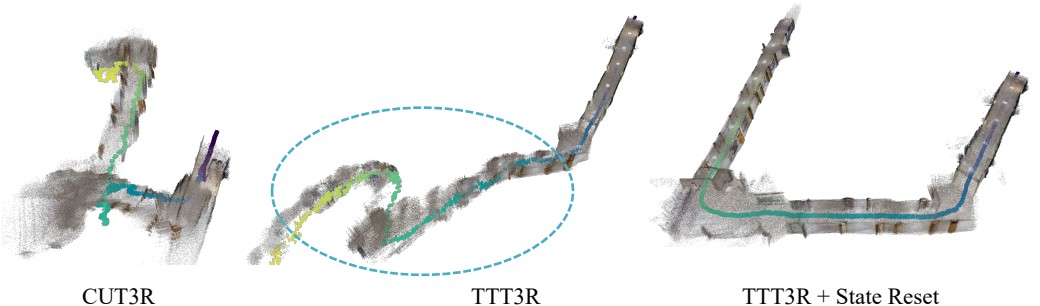

| CUT3R | TTT3R | TTT3R + State Reset |

Figure 14: **State Reset for sequences beyond 1000 frames.** As discussed in the limitation subsection, TTT3R only mitigates but does not fully resolve state forgetting, resulting in failure beyond 1000 frames. For visualization demonstrations exceeding 1000 frames (Fig. 1 and Fig. 15), we further augment TTT3R with a State Reset mechanism: the state is reset every 100 frames, and the global metric camera pose is used as a cue to align the resulting chunks. Note that, for simplicity, we **do not** employ State Reset in any quantitative experiments and analyses in the main paper. State Reset is applied only in Fig. 1 and Fig. 15 (where we visualize the final outcome, and also augment CUT3R with State Reset to enable a fair and intuitive comparison), and Fig. 13 (to report quantitative improvements achievable with State Reset).

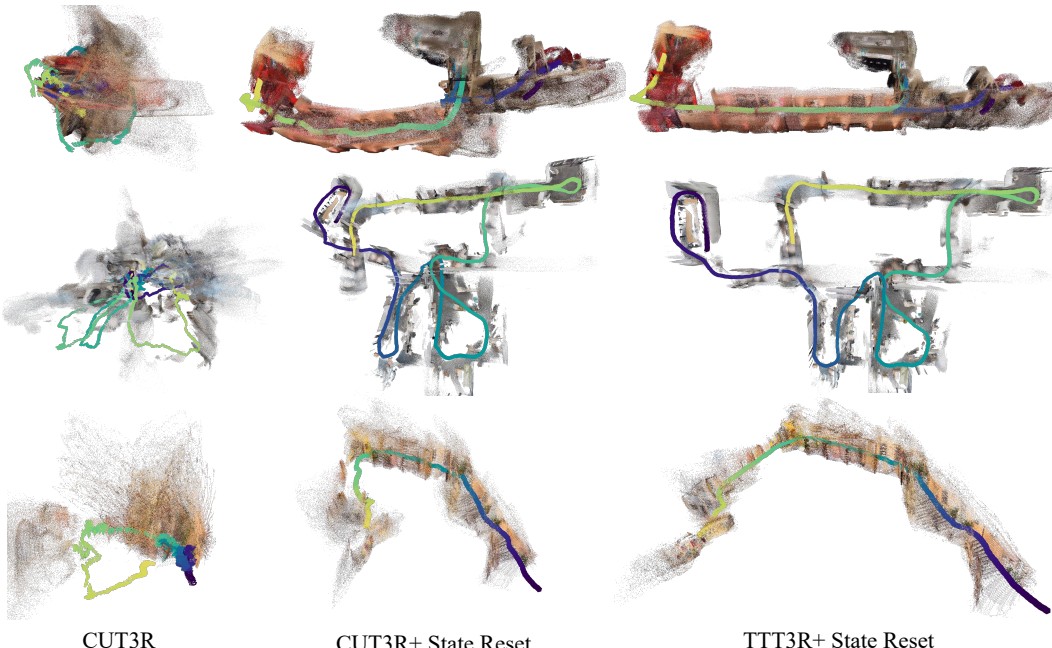

| CUT3R | CUT3R+ State Reset | TTT3R+ State Reset |

Figure 15: **In-the-wild Video Reconstruction - Sequences beyond 1000 frames.** While CUT3R + State Reset still suffers from drifting, TTT3R + State Reset enables robust long-sequence 3D reconstruction beyond 1000 frames. The TTT3R + State Reset is performed in the forward pass without any optimization, making it a plug-and-play solution that preserves CUT3R's inference speed and memory footprint.

## C    MORE RESULTS

### C.1    EXPERIMENTAL SETTINGS

We present two experimental settings: (1) long sequence evaluation, to compare TTT3R with online methods that could handle hundreds of images, which is a challenging setting by measuring the state capability to memorize entire sequences, rather than short video clips; and (2) short sequence evaluation, to compare the performance of our method to a wide range of baselines (since these baselines, hindered by out-of-memory, are infeasible to handle long sequences). Please refer to the main paper for the long-sequence results. We report the short-sequence evaluation in the following section of the supplement.

| Method | Online | Sintel (50 frames) | | | TUM-dynamics (90 frames) | | | ScanNet (90 frames) | | |
|---|---|---|---|---|---|---|---|---|---|---|
| | | ATE ↓ | RPE trans ↓ | RPE rot ↓ | ATE ↓ | RPE trans ↓ | RPE rot ↓ | ATE ↓ | RPE trans ↓ | RPE rot ↓ |
| Robust-CVD [39] | ✗ | 0.360 | 0.154 | 3.443 | 0.153 | 0.026 | 3.528 | 0.227 | 0.064 | 7.374 |
| CasualSAM [104] | ✗ | 0.141 | **0.035** | 0.615 | 0.071 | **0.010** | 1.712 | 0.158 | 0.034 | 1.618 |
| DUSt3R [91] | ✗ | 0.417 | 0.250 | 5.796 | 0.083 | 0.017 | 3.567 | 0.081 | 0.028 | 0.784 |
| MASt3R [42] | ✗ | 0.185 | 0.060 | 1.496 | 0.038 | 0.012 | 0.448 | 0.078 | 0.020 | 0.475 |
| MonST3R [102] | ✗ | 0.111 | 0.044 | 0.869 | 0.098 | 0.019 | 0.935 | 0.077 | 0.018 | 0.529 |
| Easi3R [16] | ✗ | **0.110** | 0.042 | 0.758 | 0.105 | 0.022 | 1.064 | 0.061 | 0.017 | 0.525 |
| AETHER[78] | ✗ | 0.189 | 0.054 | 0.694 | 0.092 | 0.012 | 1.106 | 0.176 | 0.028 | 1.204 |
| VGGT [87] | ✗ | 0.172 | 0.062 | **0.471** | **0.012** | **0.010** | **0.310** | **0.035** | **0.015** | **0.377** |
| Spann3R [85] | ✓ | 0.329 | 0.110 | 4.471 | 0.056 | 0.021 | 0.591 | 0.096 | 0.023 | 0.661 |
| CUT3R [89] | ✓ | 0.213 | 0.066 | 0.621 | 0.046 | 0.015 | 0.473 | 0.099 | 0.022 | 0.600 |
| Point3R [95] | ✓ | 0.351 | 0.128 | 1.822 | 0.075 | 0.029 | 0.642 | 0.106 | 0.035 | 1.946 |
| StreamVGGT [106] | ✓ | 0.251 | 0.149 | 1.894 | 0.061 | 0.033 | 3.209 | 0.161 | 0.057 | 3.647 |
| STream3R [40] | ✓ | 0.213 | 0.076 | 0.868 | **0.026** | 0.013 | **0.330** | 0.052 | 0.021 | 0.850 |
| **TTT3R** | ✓ | **0.201** | **0.063** | **0.617** | 0.028 | **0.012** | 0.379 | 0.064 | 0.021 | **0.592** |

Table 6: **Evaluation on Camera Pose Estimation- Short Sequence** on Sintel [13], TUM-dynamics [71], and ScanNet [20] datasets. TTT3R achieves the best overall performance among online methods, while its accuracy has not yet matched strong offline methods (e.g., VGGT), where full attention — despite being slower and more memory-demanding — preserves the entire history context.

## C.2 BASELINES

We first compare TTT3R with pairwise 3D reconstruction foundation models, including DUSt3R [91], MASt3R [42], MonST3R [102], and Easi3R [16], which takes a pair of views as input and requires an extra global alignment stage to consolidate the pairwise predictions. We also compare with AETHER [78] and VGGT [87], which could predict all pointmaps simultaneously, without the need for post-processing, leading to state-of-the-art 3D point and camera pose reconstruction. However, relying on global alignment or full attention limits all the aforementioned methods to handling only short image sequences, in an offline reconstruction manner, where it needs to rerun inference of all images whenever a new frame arrives. For online methods, we compare TTT3R with Spann3R [85] and CUT3R [89], which operates online with RNN-based architectures and could handle streaming images on the fly. For concurrent works that are most similar to our method, we compare TTT3R with Point3R [95], StreamVGGT [106] and STream3R [40], which aim to extend CUT3R and VGGT to handle long image sequences, but take a different approach by fine-tuning CUT3R and VGGT with explicit pointmap memory and KV cache as state representation, respectively. Unlike these works, our method introduce sequence modeling as a general framework and reformulate CUT3R from the Test-Time Training (TTT) perspective. As a result, TTT3R achieves online associative recall by deriving a closed-form update rule, without requiring fine-tuning CUT3R or training extra parameterized components.

## C.3 CAMERA POSE ESTIMATION

Following prior works [89, 102], we evaluate camera pose estimation accuracy on Sintel [13], TUM dynamics [71], and ScanNet [20] datasets. We use standard error metrics: Absolute Translation Error (ATE), Relative Translation Error (RTE), and Relative Rotation Error (RRE), after applying the Sim(3) alignment [81] on the estimated camera trajectory to the ground truth. We compare with both 3D reconstruction foundation models and prior per-sequence optimize approaches, such as RobustCVD [39] and CasualSAM [104], which jointly optimize camera parameters and dense depth maps to fit each sequence.

The results of the long-sequence evaluation are presented in Figure 7. Since many baselines can only process short sequences due to out-of-memory, we also show the short sequence evaluation in Table 6, and separately highlight the leading approaches for methods that operate offline (i.e., require additional optimization or global attention) and those that do not (i.e., online). Although a gap persists between offline and online methods, our approach achieves the best overall performance among online methods, particularly in TUM-dynamics and ScanNet datasets.

We show qualitative comparisons of the estimation of the camera trajectory in Figure 16. TTT3R demonstrates a more accurate and robust camera pose estimation over CUT3R, effectively leveraging

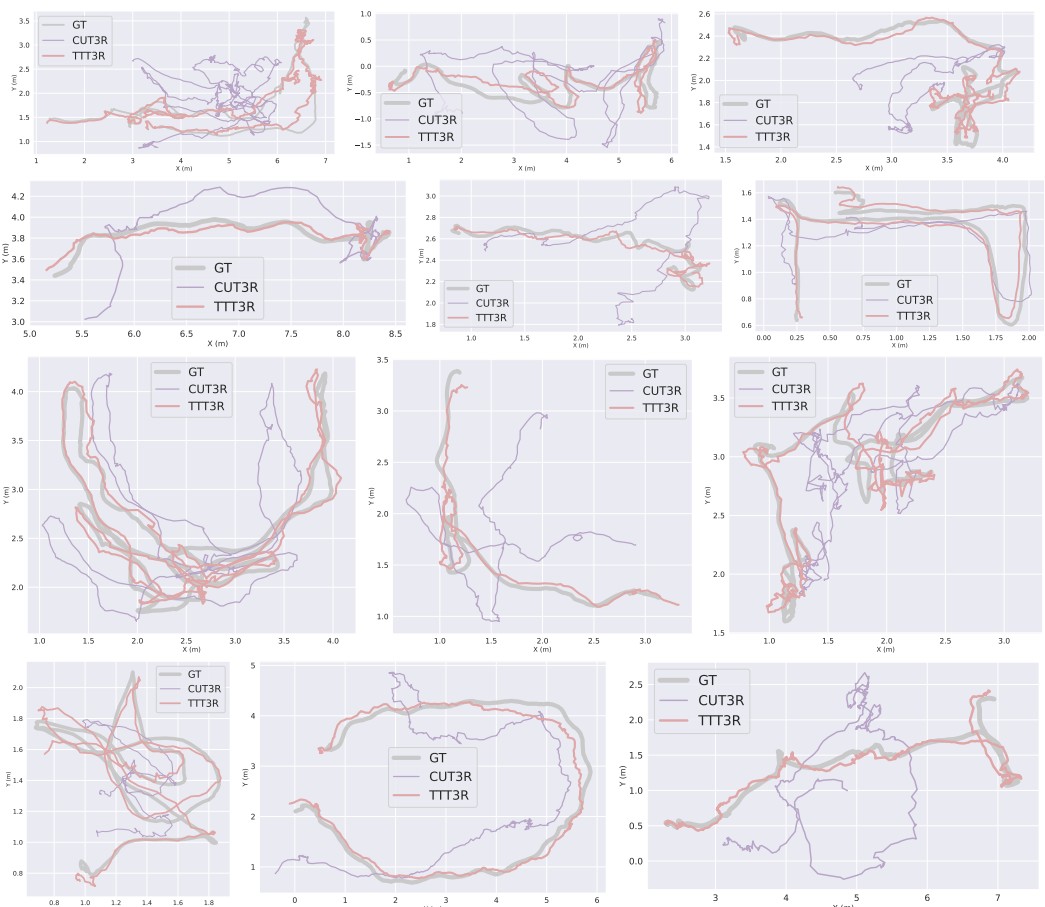

Figure 16: **Visualization of Estimated Camera Trajectories – Long Sequence.** The trajectories are plotted along the two axes with the highest variance to capture the most significant motion. Our estimated camera trajectory ● TTT3R deviates less from the ground truth ● GT compared to the baseline ● CUT3R.

the inherent knowledge of the 3D reconstruction foundation model with just a few lines of plug-and-play code by proposing a general sequence modeling formulation and deriving novel state transition rule of CUT3R from a TTT pespective.

## C.4 VIDEO DEPTH ESTIMATION

Following common practice [89, 102], we evaluate video depth estimation on KITTI [33], Sintel [13], and Bonn [55] datasets covering dynamic and static, indoor and outdoor scenes. We use absolute relative error (Abs Rel) and $\delta < 1.25$ (percentage of predicted depths within a 1.25-factor of true depth) as metrics. Video depth estimation evaluates per-frame depth quality and inter-frame depth consistency by aligning predicted depth maps to ground truth using a per-sequence scale, which measures the relative depth accuracy. For methods that predict metric pointmaps (i.e., outputs in meters with absolute scale), we also report results without scale alignment, evaluating predictions directly in metric units to assess absolute-scale accuracy.

Figure 8 presents the quantitative comparison between our method and the online baselines. Our approach delivers the best overall performance without ANY fine-tuning. The results in Table 7 show that even for short sequences, our method still achieves state-of-the-art or competitive performance in online methods, leading in KITTI dataset for both metric and scale-invariant evaluations and ranking one or two in Sintel and Bonn datasets.

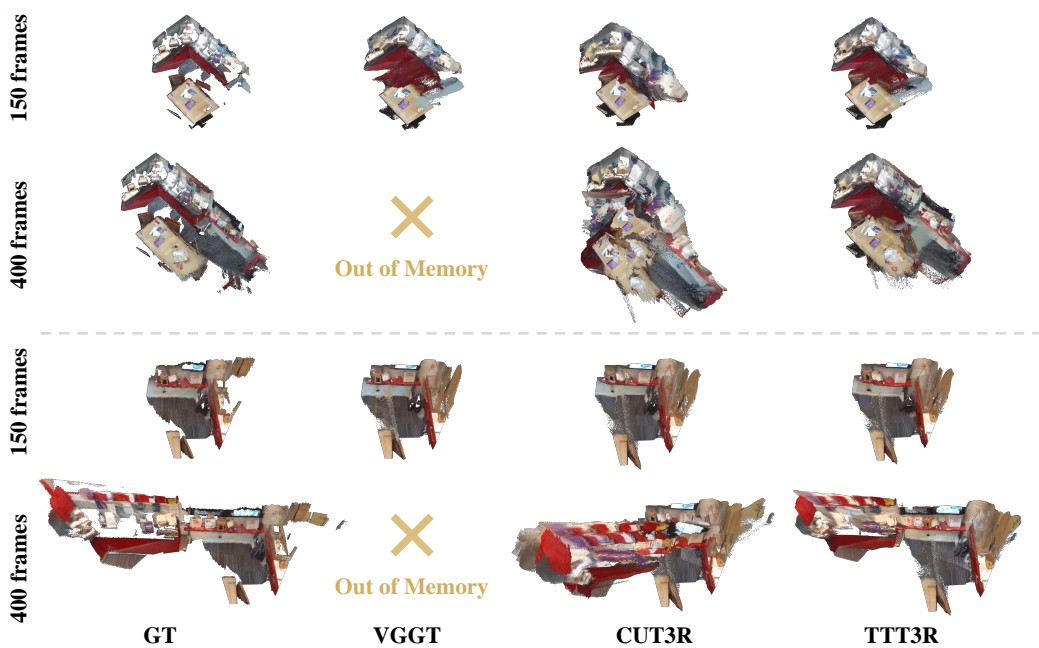

Figure 17: **Qualitative Results of 3D Reconstruction.** TTT3R improves length generalization over CUT3R while preserving its speed and memory efficiency. Offline methods (e.g., VGGT) achieve accurate reconstruction on short sequences (150 frames) but fail on longer sequences (400 frames) due to memory constraints.

| Alignment | Method | Online | Sintel (50 frames) | | BONN (110 frames) | | KITTI (110 frames) | |
|---|---|---|---|---|---|---|---|---|
| | | | Abs Rel ↓ | $\delta<1.25$ ↑ | Abs Rel ↓ | $\delta<1.25$ ↑ | Abs Rel ↓ | $\delta<1.25$ ↑ |
| Per-sequence Scale | DUSt3R [91] | ✗ | 0.656 | 45.2 | 0.155 | 83.3 | 0.144 | 81.3 |
| | MASt3R [42] | ✗ | 0.641 | 43.9 | 0.252 | 70.1 | 0.183 | 74.5 |
| | MonST3R [102] | ✗ | 0.378 | 55.8 | 0.067 | 96.3 | 0.168 | 74.4 |
| | Easi3R [16] | ✗ | 0.377 | 55.9 | 0.059 | 97.0 | 0.102 | 91.2 |
| | AETHER [78] | ✗ | 0.324 | 50.2 | 0.273 | 59.4 | **0.056** | **97.8** |
| | VGGT [87] | ✗ | **0.287** | **66.1** | **0.055** | 97.1 | 0.070 | 96.5 |
| | Spann3R [85] | ✓ | 0.622 | 42.6 | 0.144 | 81.3 | 0.198 | 73.7 |
| | CUT3R [89] | ✓ | 0.421 | 47.9 | 0.078 | 93.7 | 0.118 | 88.1 |
| | Point3R [95] | ✓ | 0.452 | 48.9 | 0.060 | 96.0 | 0.136 | 84.2 |
| | StreamVGGT [106] | ✓ | **0.323** | **65.7** | 0.059 | **97.2** | 0.173 | 72.1 |
| | STream3R [40] | ✓ | 0.478 | 51.1 | 0.075 | 94.1 | 0.116 | 89.6 |
| | **TTT3R** | ✓ | 0.404 | 50.0 | 0.068 | 95.4 | **0.113** | 90.4 |
| Metric Scale | MASt3R [42] | ✗ | 1.022 | 14.3 | 0.272 | 70.6 | 0.467 | 15.2 |
| | CUT3R [89] | ✓ | 1.029 | 23.8 | 0.103 | 88.5 | 0.122 | 85.5 |
| | Point3R [95] | ✓ | **0.777** | 17.1 | 0.137 | **94.7** | 0.191 | 73.8 |
| | STream3R [40] | ✓ | 1.041 | 21.0 | **0.084** | 94.4 | 0.234 | 57.6 |
| | **TTT3R** | ✓ | 0.977 | **24.5** | 0.090 | 94.2 | **0.110** | **89.1** |

Table 7: **Evaluation of Video Depth Estimation - Short Sequence**. We report scale-invariant relative depth (aligned by a per-sequence scale) and metric scale absolute depth accuracy on Sintel [13], Bonn [55], and KITTI [33] datasets. TTT3R achieves state-of-the-art or competitive performance among online methods, leading in KITTI for both metric and scale-invariant evaluations and ranking first or second in Sintel and Bonn.

## C.5 3D RECONSTRUCTION

The results are presented in Figure 18, Figure 15 and Figure 17. TTT3R supports both video sequences and sparse photo collections, across static and dynamic scenes, and performs online 3D reconstruction by estimating camera parameters and dense geometry for each incoming frame. TTT3R is a simple modification to CUT3R that improves length generalization via a closed-form state update, enabling robust long-sequence 3D reconstruction. The update is performed in the forward pass without any model fine-tuning, making it a plug-and-play solution, while preserving CUT3R's inference speed and memory footprint and operating online at realtime and only cost 6GB GPU memory.

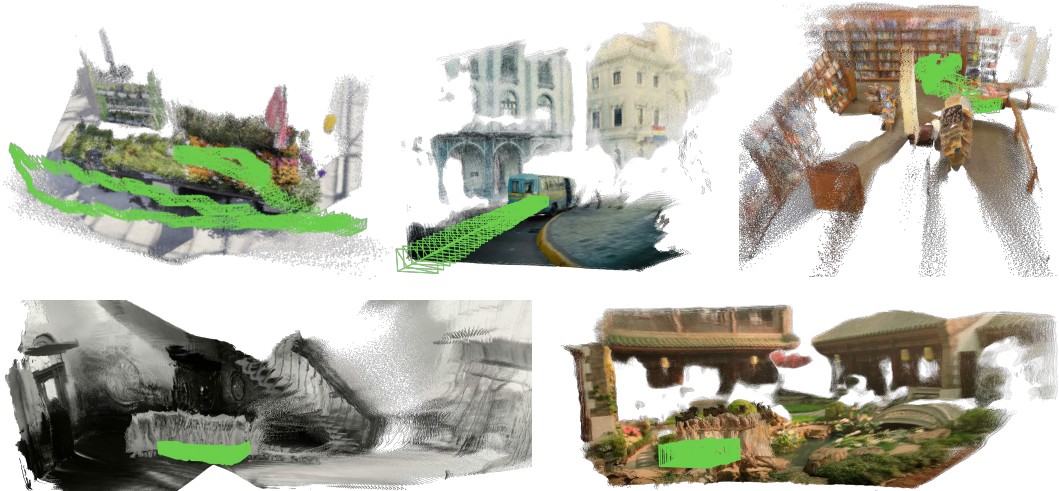

Figure 18: **In-the-Wild Video Reconstruction - Short Sequence.** TTT3R performs online 3D reconstruction by estimating camera parameters and dense geometry for each incoming image. It supports varying-length image inputs, either video streams or sparse photo collections, across both static and dynamic scenes.

# D    USE OF LARGE LANGUAGE MODELS

We used a large language model to assist with copy editing—grammar checking, wording suggestions, and minor style and clarity improvements—after the scientific content, methodology, analyses, and conclusions had been written by the authors.

