# OpenReview forum: "TTT3R: 3D Reconstruction as Test-Time Training"
_ICLR.cc/2026/Conference — ICLR 2026 Poster_

### Official Review · Reviewer_6nKa · 2025-10-29

**Soundness:** 4
**Presentation:** 4
**Contribution:** 3
**Rating:** 6
**Confidence:** 5

**Summary:**

This paper proposes TTT3R, a test-time training framework for online 3D reconstruction tasks, including depth estimation and camera pose estimation from long video sequences. While prior offline reconstruction models (e.g., VGGT, Point3R) often run out of memory when processing long sequences, TTT3R offers a training-free and memory-efficient solution. Built upon the recurrent architecture of CUT3R, it introduces a confidence-guided state update rule that adaptively balances memory retention and adaptation to new observations, leading to improved performance and generalization on long video sequences.

**Strengths:**

1. The idea is simple, intuitive, and well-motivated, reinterpreting 3D reconstruction as a test-time training problem and providing a fresh, elegant, and training-free plug-and-play solution that requires no additional parameters or fine-tuning, making it easy to integrate into existing 3D reconstruction pipelines.

2. The results are clearly better than previous online methods in long sequences, where most offline methods run out of memory.

**Weaknesses:**

1. The contribution is somewhat incremental, as the proposed method mainly builds upon CUT3R with a modified state update rule. Since TTT3R is presented as a general training-free framework, it would be valuable to include or discuss additional baselines beyond CUT3R to better demonstrate generality.

2. The performance is still lower than offline methods such as VGGT on short sequences, indicating that the proposed test-time mechanism, while efficient, does not fully close the performance gap with global-attention architectures.

3. It would also strengthen the paper to compare or at least discuss more recent approaches, such as MegaSaM, VIPE, VGGT-SLAM, and VGGT-Long. Even though some of these may not yet be peer-reviewed, acknowledging them would help position this work within the rapidly evolving landscape of long-sequence 3D reconstruction.

**Questions:**

NA

---

> ### Author Response · Authors · 2025-11-30
> **Official Response by Authors (1/2)**
>
> We appreciate the feedback. We revise the manuscript and address each point below:
>
> > **Q1: Additional baselines. “Since TTT3R is presented as a general training-free framework, it would be valuable to include or discuss additional baselines beyond CUT3R to better demonstrate generality.”**
>
> Thank you for these valuable comments. We expand the discussion to include a broader range of reconstruction foundation models in the revised manuscript `L.223–238`.
>
> **1\. Generality of the Methodology** While TTT3R is evaluated on CUT3R, we clarify that our analysis of the **cross-attention based state update rule** is a general finding, not limited to a single architecture. It offers an option for enhancing long-sequence generalization in any transformer-based reconstruction model that utilizes latent state representations.
>
> **2\. Broader Context** Recent advancements in recurrent architectures (e.g., Titans) suggest that improved state update mechanisms are important for processing long sequences. Our work investigates this direction specifically within 3D reconstruction. We hope our findings regarding memory stability and efficiency will serve as a useful reference for future research aiming to improve long-horizon performance.
>
> ---
>
> > **Q2: Performance gap with the offline methods. “The performance is still lower than offline methods such as VGGT on short sequences, indicating that the proposed test-time mechanism, while efficient, does not fully close the performance gap with global-attention architectures.”**
>
> We acknowledge this limitation and include a deeper discussion in the revised manuscript `L.525–539`.
>
> **1\. The Efficiency-Accuracy Trade-off** We clarify that the performance gap stems from the fundamental difference in computational models:
>
> * **VGGT (Offline):** Relies on global attention, preserving the entire history context. While more accurate, it is computationally intensive and memory-demanding ($O(N^2)$), making it unsuitable for long streaming sequences.
> * **TTT3R (Online):** Designed for linear complexity ($O(N)$) and constant memory ($O(1)$) . While it mitigates state forgetting, the compression of history into a fixed-size state incurs some information loss compared to full-attention retention.
>
> **2\. The "Unexplored States" Hypothesis** We attribute the remaining gap to the "unexplored states" hypothesis. Models trained on short contexts may fail to generalize to longer sequences because the recurrent updates drive the hidden state into out-of-distribution regions not encountered during training.
>
> **3\. Closing the Gap: TTT3R \+ State Reset** To address this while maintaining efficiency, we explore a **TTT3R \+ State Reset** variant (detailed in `Appendix B` and `L.1026–1122`).
>
> * By periodically resetting the state to its initial value, we effectively prevent state drift.
> * The resulting chunks are aligned using global metric poses without additional optimization.
> * This offers a plug-and-play solution that significantly improves accuracy while retaining the inference speed and memory efficiency of the recurrent architecture.
>
> While TTT3R trades some accuracy for efficiency compared to offline methods, it provides a critical capability—**linear-time processing of indefinite sequences**—that global attention methods cannot support. The Reset variant further bridges this gap, offering a practical sweet spot between accuracy and speed.

---

> ### Author Response · Authors · 2025-11-30
> **Official Response by Authors (2/2)**
>
> > **Q3: More recent approaches. “It would also strengthen the paper to compare or at least discuss more recent approaches, such as MegaSaM, VIPE, VGGT-SLAM, and VGGT-Long. Even though some of these may not yet be peer-reviewed, acknowledging them would help position this work within the rapidly evolving landscape of long-sequence 3D reconstruction.”**
>
> Thank you for this helpful suggestion. We agree that discussing these recent approaches is vital for positioning our work within the rapidly evolving landscape.
>
> **1\. Discussion of Emerging Approaches** `L.117` We incorporate a detailed discussion of the suggested works (MegaSaM, VIPE, VGGT-SLAM, VGGT-Long) in the revised manuscript:
>
> “Recent work, such as MegaSaM and VIPE, has demonstrated progress in adapting traditional SLAM paradigms to dynamic scenes by integrating semantic segmentation, optical flows, and geometric constraints. Concurrently, methods like VGGT-SLAM and VGGT-Long seek improved robustness by integrating learned front-ends. However, these methods require iterative optimization based on off-the-shelf estimation, where synchronization barriers often lead to cumulative errors and high computational overhead. This reliance hinders real-time online inference and learning scalability (e.g., the 'tabula rasa' blank slate limitation).”
>
> **Our Positioning:** In contrast, TTT3R focuses on an efficient, optimization-free recurrent update, prioritizing online speed and constant memory usage, featuring data-driven feed-forward models with generalizable priors.
>
> **2\. Additional Comparison with State-of-the-Art (STream3R)** To further demonstrate our competitiveness, we explicitly compare TTT3R with **STream3R** (a very recent approach extending VGGT for long sequences via KV caching) in the revised manuscript (`Table 6`, `Table 7`, `L.1163-1166`, `L.1145`, `L.1276`, `L.1281`).
>
> As shown in the tables below, TTT3R achieves comparable or superior accuracy to STream3R while maintaining significantly lower resource usage.
>
> | Pose | Sintel |  |  | TUM-D |  |  | ScanNet |  |  |
> | :---- | :---- | :---- | :---- | :---- | :---- | :---- | :---- | :---- | :---- |
> |  | ATE $\\downarrow$ | RPE trans $\\downarrow$ | RPE rot $\\downarrow$ | ATE $\\downarrow$ | RPE trans $\\downarrow$ | RPE rot $\\downarrow$ | ATE $\\downarrow$ | RPE trans $\\downarrow$ | RPE rot $\\downarrow$ |
> | STream3R | 0.213 | 0.076 | 0.868 | **0.026** | 0.013 | **0.330** | **0.052** | **0.021** | 0.850 |
> | TTT3R | **0.201** | **0.063** | **0.617** | 0.028 | **0.012** | 0.379 | 0.064 | **0.021** | **0.592** |
>
>
> | Depth | Method | Sintel |  | BONN |  | KITTI |  |
> | :---- | :---- | :---- | :---- | :---- | :---- | :---- | :---- |
> |  |  | Abs Rel $\\downarrow$ | $\\delta \< 1.25\\uparrow$ | Abs Rel $\\downarrow$ | $\\delta \< 1.25\\uparrow$ | Abs Rel $\\downarrow$ | $\\delta \< 1.25\\uparrow$ |
> | Per-sequence Scale | STream3R | 0.478 | **51.1** | 0.075 | 94.1 | 0.116 | 89.6 |
> |  | TTT3R | **0.404** | 50.0 | **0.068** | **95.4** | **0.113** | **90.4** |
> | Metric Scale | STream3R | 1.041 | 21.0 | **0.084** | **94.4** | 0.234 | 57.6 |
> |  | TTT3R | **0.977** | **24.5** | 0.090 | 94.2 | **0.110** | **89.1** |
>
> **Conclusion:** While STream3R utilizes full-attention history (suffering from high latency and memory exhaustion), **TTT3R achieves comparable accuracy while retaining consistent 6GB GPU memory usage and real-time efficiency.**

---

### Official Review · Reviewer_dBBg · 2025-10-31

**Soundness:** 3
**Presentation:** 3
**Contribution:** 2
**Rating:** 6
**Confidence:** 5

**Summary:**

TTT3R reframes 3D reconstruction from image sequences as a test-time training problem to address the limited length generalization of recurrent models like CUT3R. It introduces a confidence-guided state update rule, deriving a closed-form learning rate from cross-attention alignment scores. This training-free intervention balances historical retention with new observations, mitigating catastrophic forgetting. Evaluations show TTT3R improves performance on long sequences (e.g., 2× better pose estimation) while maintaining CUT3R's efficiency (20 FPS, 6 GB memory for thousands of frames).

**Strengths:**

The strengths of this paper lie in:
1) Good motivation, sequence-length generalization is a real failure mode for stateful online 3D models. Tackling it at test time without retraining is practical.
2) It is very interesting to see the training-free, plug-and-play intervention design. Requires no fine-tuning or extra parameters; preserves CUT3R’s O(1) memory and O(1) per-step compute while adding only a cheap per-token gate; practical and implementation-friendly.
3) We can see the obvious gains in length generalization (e.g., 2× ATE improvement over CUT3R on ScanNet/TUM-D at ≥200–1000 views; Fig. 6) without sacrificing throughput (20 FPS; Fig. 7) or memory (6 GB; Fig. 2).
4) The presentation of this paper has clear technical reformulation casts CUT3R’s recurrent state update as test-time training with fast weights; derives a closed-form per-token learning-rate gate beta_t from alignment confidence, Eq. 7, yielding an explicit update rule (Eq. 8) that mitigates softmax-induced overwriting.

**Weaknesses:**

May major concerns lie in:
This paper proposes a simple, empirical state-update rule to enhance sequence-length generalization for CUT3R. It seems to be a straightforward extension of CUT3R. Specifically,
1) From the TTT3R HTML, the core proposal for a state update appears to be a gradient step on a per-timestep loss with respect to the state, i.e., Update(S_{t-1}, X_t) = S_{t-1} − beta ∇_S L(S_{t-1}, X_t). This is a clean, causal mechanism aligned with the TTT literature but adapted to “fast weights”/state instead of parameters.
2) The update form given in TTT3R is easy to integrate with CUT3R,  that is to treat the persistent scene state as a latent variable and make a local corrective step using the current-frame loss. If the loss L uses only X_t (and not future frames), this is strictly causal and compatible with streaming.
3) CUT3R maintains a persistent state via state tokens and a learned update/readout (ViT decoders). CUT3R trains with a fixed context length and acknowledges resets of the state on single-view datasets, which suggests the usual risk of degradation when inference sequences are longer than the training horizon. A test-time state-adaptation rule that damps drift and improves long-horizon stability is therefore a relevant add-on.

Some other technical problems:
1) What is the test-time loss L(S_{t-1}, X_t)?
If it uses only self-supervised signals from a single frame (e.g., priors, regularizers) it may be weak/noisy. If it uses multiview photometric/reprojection consistency, you must ensure it’s causal (only past frames). Please make explicit exactly which losses are used at test time and why they are valid surrogates for “state correctness.”
If supervised losses are used in any evaluation, that would not be a realistic test-time condition—please clarify.
2) Stability analysis vs “framework to analyze dynamics”
The claim of “a TTT-based framework to analyze the dynamics” suggests more than proposing an update rule. Do you provide any formal analysis (e.g., linearization around a fixed point, Jacobian spectral radius of the state dynamics, contraction conditions w.r.t. beta, or Lyapunov-style arguments)?
If not, please temper the language or add at least a local linear analysis that shows how the beta step acts as a stabilizer/damping term and under what conditions it improves stability.
3) Evidence that the update improves sequence-length generalization in CUT3R
Please report metrics as a function of time t on sequences substantially longer than the training horizon (e.g., 2–4×). Plot error growth with/without TTT state updates, and show that the slope is reduced. Include both static and dynamic scenes since CUT3R handles both.
Provide ablations on: number of gradient steps per frame, beta schedule, gradient clipping, choice of loss, and whether backprop-through-time is disabled (it should be, for causality/cost).
Compare to strong non-TTT baselines: periodic state reset, EMA/shrinkage to s0, learned gating/forget gates, and “burn-in” with keyframes.

**Questions:**

Please see the weakness section.

---

> ### Author Response · Authors · 2025-11-30
> **Official Response by Authors (1/3)**
>
> We appreciate the feedback. We revise the manuscript and address each point below:
>
> > **Q1: A straightforward extension. “May major concerns lie in: This paper proposes a simple, empirical state-update rule to enhance sequence-length generalization for CUT3R. It seems to be a straightforward extension of CUT3R.” Specifically,**
> >1. From the TTT3R HTML, the core proposal for a state update appears to be a gradient step on a per-timestep loss with respect to the state, i.e., Update(S\_{t-1}, X\_t) \= S\_{t-1} − beta ∇S L(S{t-1}, X\_t). This is a clean, causal mechanism aligned with the TTT literature but adapted to “fast weights”/state instead of parameters.
> >2. The update form given in TTT3R is easy to integrate with CUT3R, that is to treat the persistent scene state as a latent variable and make a local corrective step using the current-frame loss. If the loss L uses only X\_t (and not future frames), this is strictly causal and compatible with streaming.
> >3. CUT3R maintains a persistent state via state tokens and a learned update/readout (ViT decoders). CUT3R trains with a fixed context length and acknowledges resets of the state on single-view datasets, which suggests the usual risk of degradation when inference sequences are longer than the training horizon. A test-time state-adaptation rule that damps drift and improves long-horizon stability is therefore a relevant add-on.
>
>
> Thank you for your valuable comments. While we agree that TTT3R conceptually integrates the causal state-update mechanism from TTT to enhance CUT3R, we respectfully emphasize that it is **not a trivial combination**.
>
> The core challenge—and our key innovation—lies in **how to model the learning rate $\\beta$ effectively without incurring prohibitive training costs.** A straightforward application of existing learnable gating mechanisms fails to achieve optimal performance due to the difficulty of training on long sequences.
>
> **1\. Novelty: Training-Free Closed-Form Derivation** Unlike standard TTT or RetNet/Mamba architectures that rely on learnable parameters for gating, we derive a **training-free, closed-form learning rate** directly from cross-attention alignment.
>
> To clarify this distinction, we compare our approach with standard learnable designs:
>
> * **ScalarLR (e.g., RetNet):** Single learnable scalar $\\beta \\in \\mathbb{R}^{1}$.
> * **ConditionLR (e.g., TTT, Mamba-2):** Input-dependent scalar function $\\beta\_t=\\sigma (\\ell\_{\\beta}(X\_t))\\in \\mathbb{R}^{1}$.
> * **TokenLR (e.g., GLA):** Input-dependent per-token function $\\beta\_t=\\sigma (\\ell\_{\\beta}(X\_t)) \\in \\mathbb{R}^{n \\times 1}$.
>
> **Our Contribution:** While TTT3R aligns with the high-capacity **TokenLR** paradigm ($\\beta\_t \\in \\mathbb{R}^{n \\times 1}$), we eliminate the need for auxiliary learnable networks (hyper-networks $\\ell\_{\\beta}$). Instead, we derive a training-free $\\beta\_t$ analytically from the confidence of memory retrieval: $\\beta\_t \= \\sigma ( \\sum\_{m} Q\_{S\_{t-1}} {K^{\\top}\_{X\_t}} )$.
>
> **2\. Evidence: Why "Straightforward Extension" is Insufficient** To prove that our design is superior to a "straightforward" learnable extension, we implement ScalarLR, ConditionLR, and TokenLR on top of CUT3R (freezing the backbone and training only the gates).
>
> As shown in the table below (`Table 1`, `Figure 11` and `L.891–957` in revision), simply adding a learnable gating module (even the expressive TokenLR) is **significantly less effective** than our derived update rule.
>
> | Method | Camera Pose |  | Video Depth |  |
> | :---- | :---- | :---- | :---- | :---- |
> |  | ATE $\\downarrow$ | RPE rot $\\downarrow$ | Abs Rel $\\downarrow$ | $\\delta \< 1.25\\uparrow$ |
> | CUT3R | 0.173 | 0.494 | 0.152 | 80.2 |
> | CUT3R \+ ScalarLR | 0.165 | 0.502 | 0.151 | 80.6 |
> | CUT3R \+ ConditionLR | 0.166 | 0.509 | 0.149 | 81.0 |
> | CUT3R \+ TokenLR | 0.154 | 0.497 | 0.148 | 81.5 |
> | **TTT3R** | **0.106** | **0.431** | **0.131** | **86.9** |
>
> In our observation, the performance of the learnable gating mechanisms is primarily limited by the training sequence length (i.e. 64 frames). We find that the longer the training sequence, the better the learnable gating mechanism performs. Unfortunately, training on sequences longer than 64 frames is prohibitively costly and becomes infeasible given our hardware constraints (48GB NVIDIA GPU).
>
> **Analysis:** The performance of learnable extensions is strictly limited by the training sequence length (max 64 frames in our setting). Learning to generalize to long horizons requires training on long horizons, which is computationally intractable. **Our training-free formulation bypasses this bottleneck**, achieving superior adaptivity and generalization by leveraging intrinsic geometric alignment confidence.

---

> ### Author Response · Authors · 2025-11-30
> **Official Response by Authors (2/3)**
>
> > **Q2: Loss functions. “What is the test-time loss L(S\_{t-1}, X\_t)? If it uses only self-supervised signals from a single frame (e.g., priors, regularizers) it may be weak/noisy. If it uses multiview photometric/reprojection consistency, you must ensure it’s causal (only past frames). Please make explicit exactly which losses are used at test time and why they are valid surrogates for “state correctness.” If supervised losses are used in any evaluation, that would not be a realistic test-time condition—please clarify.”**
>
> We would like to clarify that TTT3R does not perform backpropagation on a supervised test-time loss during inference. Instead, the gradient is intrinsic to the network's forward pass, derived directly from the fixed parameters of the trained model.
>
> **1\. Mechanism: Gradient via Explainable Cross-Attention** Specifically, in our reformulation, the state update **is a gradient step** that is implemented efficiently via **explainable cross-attention layers**.
>
> * These layers implicitly capture **causal geometric consistency constraints** as self-supervised signals.
> * The "gradient" computation is functionally performed by the attention mechanism during the forward pass, without explicit differentiation.
>
> **2\. Validity: Robust Surrogates via Meta-Learning** To address the concern about "weak/noisy" signals: The properties of these updates—effectively the **gradient** and **learning rate**—are **meta-learned end-to-end** on the training set to minimize the point reconstruction loss.
>
> * This ensures that the implicit signals serve as **robust, data-driven surrogates** for state correctness.
> * This is strictly superior to relying on weak or noisy handcrafted priors or regularizers.
>
> **3\. Strict Causality for Streaming** Consequently, during inference and evaluation, **no supervised losses or ground-truth labels are required**. The model relies solely on these implicit self-supervised signals processed causally from past frames, ensuring the method is **strictly causal** and valid for streaming settings.
>
> ---
>
> > **Q3: “The claim of “a TTT-based framework to analyze the dynamics” suggests more than proposing an update rule. Do you provide any formal analysis (e.g., linearization around a fixed point, Jacobian spectral radius of the state dynamics, contraction conditions w.r.t. beta, or Lyapunov-style arguments)? If not, please temper the language or add at least a local linear analysis that shows how the beta step acts as a stabilizer/damping term and under what conditions it improves stability.”**
>
> Thank you for this constructive feedback. To ensure terminological precision and strictly align with our methodology, we refine the language in the revised manuscript `L.104`.
>
> Following your suggestion, we update the phrasing from "analyze the dynamics" to **"analyze the behavior"**. This revision more accurately reflects our contribution.

---

> ### Author Response · Authors · 2025-11-30
> **Official Response by Authors (3/3)**
>
> > **Q4: Plots for error growth. “Evidence that the update improves sequence-length generalization in CUT3R Please report metrics as a function of time t on sequences substantially longer than the training horizon (e.g., 2–4×). Plot error growth with/without TTT state updates, and show that the slope is reduced. Include both static and dynamic scenes since CUT3R handles both. Provide ablations on: number of gradient steps per frame, beta schedule, gradient clipping, choice of loss, and whether backprop-through-time is disabled (it should be, for causality/cost). Compare to strong non-TTT baselines: periodic state reset, EMA/shrinkage to s0, learned gating/forget gates, and “burn-in” with keyframes.”**
>
> Thank you for this insightful feedback. We address all the requested points below.
>
> **1\. Evidence of Generalization** `Figures 7–9` We provide concrete evidence that TTT3R significantly improves sequence-length generalization. We plot error growth metrics as a function of time on sequences of **1,000 frames** (approximately $15\\times$ the training horizon of 64 frames).
>
> * **Slope Reduction:** TTT3R reduces the error accumulation rate (slope) by approximately **$2\\times$** in camera pose estimation.
> * **Robustness:** This evaluation covers both static (ScanNet, 7-Scenes) and dynamic (TUM-D, Bonn, KITTI) scenes.
>
>
> **2\. Implementation Details & Ablations**
>
> * **Gradient Steps:** We set the number of gradient steps per frame to $k=1$. This is a design choice to ensure online efficiency, as additional steps would increase latency in streaming settings.
> * **Causality:** Backpropagation-through-time (BPTT) is **strictly disabled** to adhere to the causal nature of the task and to maintain constant memory usage.
> * **Schedule & Clipping:** We do not employ a manual $\\beta$ schedule or gradient clipping. As detailed in Q1, our formulation naturally derives an adaptive, input-dependent learning rate, which inherently regulates the update magnitude.
>
> **3\. Comparison with Non-TTT Baselines** (`Table 2`, `Table 3`, `Table 4`, `Table 5`, `Figure 12`, `L.972-1025`) We implement three strong baselines: **Periodic Reset**, **Exponential Moving Average (EMA)**, and **Burn-In**. To ensure a fair comparison, we first perform a grid search to find the optimal hyperparameters for each baseline:
>
> **(1) CUT3R \+ Reset** This mechanism performs a periodic state reset, where the state is reset to its initial value every $n$ frames. The resulting state chunks are then globally aligned using metric camera poses. We ablate the choice of the reset period $n \\in {50, 100, 150}$. As shown below, $n=100$ yields the best results.
>
> | Baselines | ATE ↓ | RPE trans ↓ | RPE rot ↓ |
> | :---- | :---- | :---- | :---- |
> | Reset 50 | 0.129 | 0.007 | 0.406 |
> | Reset 100 | **0.126** | **0.007** | **0.403** |
> | Reset 150 | 0.145 | 0.008 | 0.416 |
>
> **(2) CUT3R \+ EMA** This method introduces shrinkage towards the initial state $S\_0$ using an Exponential Moving Average (EMA): $S\_{t} \= (1 \- \\alpha) S\_{t-1} \+ \\alpha S\_{0}$. We ablate the shrinkage rate $\\alpha \\in {0.0001, 0.001, 0.01}$. As shown below, $\\alpha=0.001$ provides the optimal performance.
>
> | Baselines | ATE ↓ | RPE trans ↓ | RPE rot ↓ |
> | :---- | :---- | :---- | :---- |
> | EMA 0.0001 | 0.169 | 0.007 | 0.574 |
> | EMA 0.001 | **0.164** | **0.007** | **0.525** |
> | EMA 0.01 | 0.191 | 0.009 | 0.673 |
>
> **(3) CUT3R \+ BurnIn** The burn-in mechanism updates the state exclusively using keyframes. We ablate the keyframe interval $n \\in {50, 100, 150}$ frames. As shown below, an interval of $n=100$ yields the best results.
>
> | Baselines | ATE ↓ | RPE trans ↓ | RPE rot ↓ |
> | :---- | :---- | :---- | :---- |
> | BurnIn 50 | 0.150 | 0.0384 | 3.784 |
> | BurnIn 100 | **0.144** | **0.0309** | **3.093** |
> | BurnIn 150 | 0.156 | 0.0360 | 3.693 |
>
> **4\. Final Comparison & Orthogonality** As shown in the table below and `Figure 12` in the revised manuscript, TTT3R significantly outperforms all optimized non-TTT baselines.
>
> | Method | Camera Pose |  | Video Depth |  |
> | :---- | :---- | :---- | :---- | :---- |
> |  | ATE $\\downarrow$ | RPE rot $\\downarrow$ | Abs Rel $\\downarrow$ | $\\delta \< 1.25\\uparrow$ |
> | CUT3R | 0.173 | 0.494 | 0.152 | 80.2 |
> | CUT3R \+ Reset | 0.126 | 0.403 | 0.128 | 84.9 |
> | CUT3R \+ EMA | 0.164 | 0.525 | 0.150 | 80.7 |
> | CUT3R \+ BurnIn | 0.144 | 3.093 | 0.151 | 80.2 |
> | **TTT3R** | 0.106 | 0.431 | 0.131 | 86.9 |
> | **TTT3R \+ Reset** | **0.093** | **0.375** | **0.115** | **88.5** |
>
> **Complementary Benefits:** Interestingly, we observe that the Reset mechanism (which prevents state overfitting) is **orthogonal** to our contribution. Integrating Reset ($n=100$) into TTT3R yields the best overall performance (TTT3R \+ Reset), further validating the flexibility of our framework. Detailed analysis of TTT3R \+ Reset is included in Section B (`Figure 13`, `Figure 14`, `Figure 15` and `L.1026-1122`).

---

### Official Review · Reviewer_jkPo · 2025-11-01

**Soundness:** 3
**Presentation:** 2
**Contribution:** 3
**Rating:** 6
**Confidence:** 3

**Summary:**

This paper, TTT3R, introduces a training-free method to improve long-sequence generalization in recent recurrent 3R models (in particular, CUT3R). The task is feed-forward point-cloud reconstruction from a sequence of RGB frames. Prior work on this task includes DUSt3R, VGGT, CUT3R, et al. For long video, feed-forward reconstruction, recurrent transformers like CUT3R are generally considered competitive.

The contribution can be summarized in one sentence: rather than updating every state token as in CUT3R, TTT3R redesigns a per-token update weight (a “learning rate” in the paper’s test-time-training view) so that only those state tokens that are highly relevant to the current observations receive large updates.

In summary, the paper presents a training-free, neat idea with useful insights into how to weight state updates. The experiments show that this approach improves CUT3R’s generalization on long sequences. Nice idea, and nice results.

That said, I think the presentation would benefit from revision. Personally, I wouldn’t frame the method as “test-time training.” The same idea could be described more directly and precisely without introducing new terminology. See details below.

Overall, this is a good paper with a somewhat complex presentation. I would consider accepting it if the authors rephrase certain terms.

**Strengths:**

- A neat and elegant idea. Only a small amount of code modification is needed to alleviate the forgetting problem of CUT3R on long-sequence inputs.
- The results are impressive. Figures 7, 8, and 9 clearly show that this method achieves significantly better reconstruction quality than CUT3R.
- The visualizations are well done. Thank you for including Figure 10 and the supplementary webpage.

**Weaknesses:**

The main weakness lies in the presentation.

I would not frame the method as “test-time training.” The same idea could be described more directly and precisely without introducing new terminology. For example, I find Equation (5) unhelpful in conveying the main idea. It introduces gradient notation for a reconstruction error, which requires readers to spend time understanding the notation and the context of TTT. Once readers parse what TTT and Equation (5) mean, the paper essentially presents Equation (6), which is not really related to TTT. I would describe the approach instead as “per-token adaptive weighting of state updates,” which is much more intuitive.

Perhaps I misunderstood something, but in Equation (6), is it truly necessary to introduce the gradient notation?

**Questions:**

- I am very interested to know whether the authors have tried applying this new updating rule to fine-tune the CUT3R model. Could the performance be further improved?
- The authors present an insightful design for the weight updating rule. Have they explored alternative approaches to weight updating? I would be interested to learn about that as well.

---

> ### Author Response · Authors · 2025-11-30
> **Official Response by Authors (1/2)**
>
> We appreciate the feedback. We revise the manuscript and address each point below:
>
> > **Q1: Information Flow. “I would not frame the method as “test-time training.” The same idea could be described more directly and precisely without introducing new terminology. For example, I find Equation (5) unhelpful in conveying the main idea. It introduces gradient notation for a reconstruction error, which requires readers to spend time understanding the notation and the context of TTT. Once readers parse what TTT and Equation (5) mean, the paper essentially presents Equation (6), which is not really related to TTT. I would describe the approach instead as “per-token adaptive weighting of state updates,” which is much more intuitive. Perhaps I misunderstood something, but in Equation (6), is it truly necessary to introduce the gradient notation?”**
>
> Thank you for this valuable feedback. We agree that while the TTT framework connects our work to broader theoretical foundations, the gradient notation can be abstract. We also agree that "per-token adaptive weighting" offers a much more intuitive physical interpretation.
>
> To address this, we revise the manuscript `L.267–313` to present this intuitive perspective alongside the TTT formulation.
>
> **1\. Intuitive Interpretation of Eq. (5)** `L.267–269` We begin with the physical meaning of the TTT components by mapping them to memory operations:
>
> “Here, the key $K\_{X\_t}$ serves as an index into the previous state $S\_{t-1}$, identifying the entries to be updated with the value $V\_{X\_t}$.  Conceptually, this mechanism treats the state as a dynamic associative memory: the key specifies **where to write**, the value specifies **what to write**, and the learning rate acts as a gating mechanism that controls the memory plasticity by **weighting** the intensity of the state update.”
>
> **2\. Clarifying Gradient Notation in Eq. (6)** `L.296–298` Building on this, we directly explain the gradient function of our TTT3R reformulation via cross-attention alignment:
>
> “Conceptually, the gradient function leverages cross-attention alignment between state query and observation key to determine **where to write**, assigning the corresponding observation value as **what to write** to each state token.”
>
> **3\. Motivation via Memory Plasticity** `L.305–308` We then discuss the forgetting issue from TTT perspective to clarify our motivation:
>
> “This forgetting reflects the structural discrepancy relative to standard TTT: the lack of a flexible learning rate (effectively, a constant $\\beta\_t \= 1.0$). Consequently, we are motivated to introduce a state update **weight** that serves as a gating mechanism to explicitly control memory plasticity.”
>
> **4\. Adopting "Adaptive Weighting" Terminology** `L.312–313` Finally, we explicitly incorporate your suggestion to frame our contribution:
>
> “Our core idea is to utilize alignment confidence between memory and observation to guide state updates. This confidence constitutes an **adaptive per-token state update weight**, serving as the per-token learning rate function in the TTT formulation.”
>
> We believe these revisions effectively bridge the gap between the general TTT formulation and the intuitive "adaptive weighting" perspective.

---

> ### Author Response · Authors · 2025-11-30
> **Official Response by Authors (2/2)**
>
> > **Q2: Fine-tuning. “I am very interested to know whether the authors have tried applying this new updating rule to fine-tune the CUT3R model. Could the performance be further improved?”**
>
> Great suggestion. We include the results in the revised manuscript `Table 1`, `Figure 11` and `L.959–971`.
>
> **1\. Experimental Setup** We fine-tune TTT3R by applying our derived update rule and confidence-guided learning rate during the training process. The training follows the original CUT3R protocol, using sequences ranging from 4 to 64 views.
>
> **2\. Results** As shown in the Table below and `Figure 11` in the revised manuscript, fine-tuning yields a **trade-off**: it further improves camera pose estimation (ATE) but leads to a slight degradation in video depth estimation.
>
> | Method | Camera Pose |  | Video Depth |  |
> | :---- | :---- | :---- | :---- | :---- |
> |  | ATE $\\downarrow$ | RPE rot $\\downarrow$ | Abs Rel $\\downarrow$ | $\\delta \< 1.25\\uparrow$ |
> | **TTT3R** | 0.106 | **0.431** | **0.131** | **86.9** |
> | **TTT3R \+ Finetune** | **0.091** | 0.434 | 0.133 | 86.3 |
>
> **3\. Analysis** We attribute this phenomenon to tuning forgetting and training length constraint:
>
> * **Global vs. Local Trade-off:** As observed in the original CUT3R work, compared to the Stage 1 model (trained on 4 views), fine-tuning on longer sequences (4-64 views) causes the model to prioritize global alignment (improving Pose) at the expense of per-view prediction accuracy (degrading Depth). This is a known issue of tuning forgetting.
> * **Sequence Length Limitation:** As shown in `Figure 7`, TTT3R shows minimal gains over CUT3R on short sequences (e.g., 64 frames). The proposed update rule offers clear benefits for long-sequence adaptation. While scaling up training sequence length could potentially improve performance, it is computationally prohibitive with current hardware.
>
> **Conclusion:** Since our method achieves length generalization without the need for expensive fine-tuning, we maintain the "training-free" TTT3R as our primary recommendation. However, fine-tuning remains a viable option for pose-centric applications or future scenarios where efficient long-sequence training becomes feasible.
>
> ---
>
> > **Q3: Alternative approaches. “The authors present an insightful design for the weight updating rule. Have they explored alternative approaches to weight updating? I would be interested to learn about that as well.”**
>
> Thank you for the positive comment on our design. To further validate our design, we explore alternative weight updating mechanisms inspired by recent advances and incorporate a comparison in `Table 1`, `Figure 11`, and `L.891–957`.
>
> **1\. Alternative Approaches Explored.** We implement three representative variants as baselines:
>
> * **ScalarLR (e.g., RetNet):** A single learnable scalar parameter shared across all updates.
> * **ConditionLR (e.g., DeltaNet, TTT, Mamba-2):** An input-dependent scalar function (conditioning on observation).
> * **TokenLR (e.g., Gated Linear Attention):** An input-dependent, per-token vector function (most similar to our dimensionality).
>
> **2\. Comparative Results** We integrate these mechanisms into CUT3R (freezing the backbone and training only the new gating modules). As shown in the table below and `Table 1` in revision, while learnable mechanisms—especially TokenLR—do improve performance, **TTT3R significantly outperforms the alternatives.**
>
> | Method | Camera Pose |  | Video Depth |  |
> | :---- | :---- | :---- | :---- | :---- |
> |  | ATE $\\downarrow$ | RPE rot $\\downarrow$ | Abs Rel $\\downarrow$ | $\\delta \< 1.25\\uparrow$ |
> | CUT3R | 0.173 | 0.494 | 0.152 | 80.2 |
> | CUT3R \+ ScalarLR | 0.165 | 0.502 | 0.151 | 80.6 |
> | CUT3R \+ ConditionLR | 0.166 | 0.509 | 0.149 | 81.0 |
> | CUT3R \+ TokenLR | 0.154 | 0.497 | 0.148 | 81.5 |
> | **TTT3R** | **0.106** | **0.431** | **0.131** | **86.9** |
>
> **3\. Why TTT3R is Superior** The performance gap highlights a critical limitation of learnable updates: they require training on very long sequences to learn effective gating policies for long-range adaptation. This is computationally prohibitive (e.g., infeasible beyond 64 frames on standard hardware).
>
> In contrast, our **training-free** approach derives the update weight analytically from alignment confidence, allowing it to generalize to long sequences without requiring expensive long-sequence training, making it both more effective and more efficient than learnable alternatives.

---

### Official Review · Reviewer_yvT7 · 2025-11-01

**Soundness:** 3
**Presentation:** 3
**Contribution:** 3
**Rating:** 6
**Confidence:** 5

**Summary:**

This paper addresses the critical issue of catastrophic forgetting and poor length generalization in recurrent 3D reconstruction foundation models. It re-frames the recurrent state update mechanism as an online, test-time training (TTT) problem. The primary contribution is TTT3R, a novel, training-free framework that derives a closed-form adaptive learning rate based on the alignment confidence between the memory state and new observations. This dynamically balances memory retention and adaptation, enabling robust and accurate 3D reconstruction and pose estimation over long image sequences without incurring additional computational or memory costs.

**Strengths:**

1. The re-interpretation of the RNN state update (specifically in CUT3R) as a test-time training process is a novel and insightful conceptual contribution. The derivation of a closed-form, confidence-guided learning rate (Eq. 7) from this perspective is an elegant solution to the problem of balancing state retention and adaptation.

2. The method is training-free and acts as a plug-and-play module for the CUT3R baseline. It achieves significant performance gains, including a 2x improvement in global pose ATE on long sequences. It consistently outperforms other online methods (CUT3R, Point3R) on multiple challenging benchmarks (ScanNet, TUM-D) while, crucially, maintaining the same low memory footprint (6GB) and real-time inference speed (20 FPS) as the baseline.

3.   The paper tackles a highly significant and practical bottleneck in the field. Length generalization is a major limiting factor for applying efficient, streaming 3D reconstruction models to real-world, long-term video sequences for applications like SLAM or AR.

**Weaknesses:**

1. Clarity of Final Update Rule: The presentation of the final, implemented update rule is confusing. Equation (8) merely restates the generic TTT formula. The actual update rule, which appears to be $S_t = S_{t-1} + \beta_t \cdot [softmax(Q_{S_{t-1}} K_{X_t}^\top) V_{X_t}]$ (where $\beta_t$ is from Eq. 7), is only implied by diagrams (Fig. 4, 5) and the analysis of CUT3R (Eq. 6). The paper would be much clearer if this final equation was explicitly stated.

2. Relation to Gating Mechanisms: The proposed confidence-guided learning rate $\beta_t$ is functionally very similar to a standard gating mechanism. The paper could be strengthened by a more thorough discussion of the conceptual and practical differences between its TTT-derived update rule and simply adding a more traditional, or even learnable, gating module to the CUT3R state update.

3. Another relevant work, Test3R, also present test-time training based on dust3r series with self-consistency guided adaptation. A discussion and comparison with this method would be necessary to better contextualize the paper's contribution.

[1] Test3R: Learning to Reconstruct 3D at Test Time, NeurIPS 2025

**Questions:**

Please see the weakness part.

---

> ### Author Response · Authors · 2025-11-30
> **Official Response by Authors (1/3)**
>
> We appreciate the feedback. We revise the manuscript and address each point below:
>
> > **Q1: Clarity of Final Update Rule. “The presentation of the final, implemented update rule is confusing. Equation (8) merely restates the generic TTT formula. The actual update rule, which appears to be $S\_t \= S\_{t-1} \+ \\beta\_t \\cdot \[ \\text{softmax}(Q\_{S\_{t-1}} K\_{X\_t}^\\top) V\_{X\_t} \]$(where $\\beta\_t$ is from Eq. 7), is only implied by diagrams (Fig. 4, 5\) and the analysis of CUT3R (Eq. 6). The paper would be much clearer if this final equation was explicitly stated.”**
>
> Thank you for pointing this out. We explicitly formulate the final update rule in the revised manuscript `L.322–329` to enhance clarity:
>
> $$
> S\_t \= S\_{t-1} \- \\sigma \\left(\\sum\_m Q\_{S\_{t-1}} K\_{X\_t}^\\top\\right) \\cdot \\left(- \\text{softmax}(Q\_{S\_{t-1}} K\_{X\_t}^\\top) V\_{X\_t}\\right)
> $$
>
> In this revision, we address the ambiguity and align the mathematical formulation with the diagrams in Figures 4 and 5\.

---

> ### Author Response · Authors · 2025-11-30
> **Official Response by Authors (2/3)**
>
> > **Q2: Relation to Gating Mechanisms. “The proposed confidence-guided learning rate $\\beta\_t$ is functionally very similar to a standard gating mechanism. The paper could be strengthened by a more thorough discussion of the conceptual and practical differences between its TTT-derived update rule and simply adding a more traditional, or even learnable, gating module to the CUT3R state update.”**
>
> Thank you for this valuable suggestion. We agree that such an ablation is valuable for bringing further insights. We include this comparison in `Table 1`, `Figure 11`, and an analysis in `L.891–957`.
>
> **1\. Conceptual Comparison with Standard Gating Mechanisms.** To analyze the relationship between our proposed confidence-guided learning rate and existing standard gating mechanisms, we compare how the learning rate $\\beta$ is modeled in recent advances:
>
> * **ScalarLR (e.g., RetNet):** The learning rate is a single learnable scalar parameter, $\\beta \\in \\mathbb{R}^{1}$. The constraint that all observations share the same learning rate inherently limits its representational flexibility.
> * **ConditionLR (e.g., DeltaNet, TTT, Mamba-2):** The learning rate is an input-dependent scalar function: $\\beta\_t=\\sigma (\\ell\_{\\beta}(X\_t))\\in \\mathbb{R}^{1}$. This conditions the learning rate on the current observation $X\_t$ but shares it across tokens.
> * **TokenLR (e.g., Gated Linear Attention):** This proposes a $\\beta\_t$ that is both input-conditioned and per-token: $\\beta\_t=\\sigma (\\ell\_{\\beta}(X\_t)) \\in \\mathbb{R}^{n \\times 1}$. This enables adaptive assignment of individual learning rates for the $n$ state tokens.
>
> **Our Approach (TTT3R):** Our method follows the **TokenLR** paradigm ($\\beta\_t \\in \\mathbb{R}^{n \\times 1}$). However, in contrast to previous mechanisms that rely on **additional learnable parameters** (e.g., a scalar in ScalarLR, or a hyper-network $\\ell\_{\\beta}(\\cdot)$ in ConditionLR/TokenLR), we derive a **training-free, closed-form** learning rate. Motivated by the explainable cross-attention mechanism, it is derived directly from the alignment confidence between state queries and observation keys: $\\beta\_t \= \\sigma ( \\sum\_{m} Q\_{S\_{t-1}} {K^{\\top}\_{X\_t}} )$. This formulation achieves per-token adaptivity **without introducing any additional parameters**, training overhead, or computational cost.
>
> **2\. Practical Differences and Effectiveness.** To analyze the effectiveness, we compare TTT3R against simply incorporating a standard learnable gating module into the CUT3R state update.
>
> **Experimental Setup:** We introduce ScalarLR, ConditionLR, and TokenLR into the CUT3R state update. We freeze the encoder, decoder, and output heads, training only the newly added gating module on sequences of length 4 to 64\.
>
> **Results:** As shown in Table below and `Figure 11` in the revised manuscript, **CUT3R \+ TokenLR** is the most effective among learnable gates. However, it still significantly underperforms **TTT3R** in both camera pose estimation and video depth estimation.
>
> | Method | Camera Pose |  | Video Depth |  |
> | :---- | :---- | :---- | :---- | :---- |
> |  | ATE $\\downarrow$ | RPE rot $\\downarrow$ | Abs Rel $\\downarrow$ | $\\delta \< 1.25\\uparrow$ |
> | CUT3R | 0.173 | 0.494 | 0.152 | 80.2 |
> | CUT3R \+ ScalarLR | 0.165 | 0.502 | 0.151 | 80.6 |
> | CUT3R \+ ConditionLR | 0.166 | 0.509 | 0.149 | 81.0 |
> | CUT3R \+ TokenLR | 0.154 | 0.497 | 0.148 | 81.5 |
> | **TTT3R** | **0.106** | **0.431** | **0.131** | **86.9** |
>
> **Analysis on Training Constraints:** We observe that the performance of learnable gating mechanisms is primarily limited by the training sequence length. While learnable gates perform better with longer sequences, training on sequences longer than 64 frames becomes prohibitively costly (exceeding the memory of a 48GB GPU). In contrast, our approach derives the learning rate directly from context, naturally capturing long-range dependencies without requiring expensive long-sequence training.

---

> ### Author Response · Authors · 2025-11-30
> **Official Response by Authors (3/3)**
>
> > **Q3: Another relevant work. “Test3R, also present test-time training based on dust3r series with self-consistency guided adaptation. A discussion and comparison with this method would be necessary to better contextualize the paper's contribution.”**
>
> We appreciate the reviewer for highlighting Test3R. We recognize it as an inspiring and significant work that pioneered the use of geometric self-consistency for adapting 3D foundation models. We incorporate a detailed discussion and citation in our revised manuscript `L.184–186`. To better contextualize our contribution, we clarify the distinct methodological paths taken by the two works:
>
> * **Test3R (Optimization-based TTA):** This method achieves impressive reconstruction quality by **fine-tuning model weights** via backpropagation to minimize a self-consistency loss. It represents a robust optimization-based approach to scene adaptation.
> * **Our Approach (Architectural TTT):** In contrast, we explore an architectural avenue. Instead of iterative optimization, we revisit previous works under  a general TTT perspective and propose to update the **hidden state (memory)** via a closed-form rule. This allows for efficient view scalability and memory retention without the computational cost of gradient-based fine-tuning.

---

### Author Response · Authors · 2025-11-30
**General Response (2/2)**

## Part II: Key Highlights from Requested Experiments

To address concerns regarding baselines, long-sequence generalization, and comparisons with recent methods, we conduct comprehensive new experiments. We summarize the three most critical findings below:

**1\. TTT3R outperforms Learnable Gating baselines (R1, R2, R3)**
Reviewers question whether a standard learnable gating mechanism (e.g., as in RetNet or Mamba) would suffice. We implement three **learnable variants: ScalarLR, ConditionLR, and TokenLR.**

**Result:** As shown in table below, simply adding learnable gates yields limited gains. **TTT3R significantly outperforms even the best learnable variant (TokenLR)**, reducing the ATE error by **\~31%**.

| Method | Description | Camera Pose (ATE$\\downarrow$) | Video Depth (Abs Rel$\\downarrow$) |
| :---- | :---- | :---- | :---- |
| CUT3R | Baseline | 0.173 | 0.152 |
| \+ ScalarLR | Learnable scalar $\\beta$ | 0.165 | 0.151 |
| \+ ConditionLR | Input-dependent $\\beta\_t$ | 0.166 | 0.149 |
| \+ TokenLR | Per-token learned $\\beta\_t$ | 0.154 | 0.148 |
| **TTT3R (Ours)** | **Training-free derived $\\beta\_t$** | **0.106 (-31%)** | **0.131** |

**Insight:** Learnable gates require training on long sequences to generalize, which is computationally prohibitive (memory constraints). In contrast, our **training-free** derived update naturally generalizes to long sequences, validating our core technical contribution.

---

&nbsp;

**2\. Superior Long-Sequence Generalization (R3)**
We evaluate TTT3R on sequences **15x longer** (1,000 frames) than the training horizon (64 frames). We compare it against **CUT3R \+ non-TTT baselines:** Exponential Moving Average (EMA), Burn-In, and Periodic Reset.

**Result:** As visualized in table below and **`Figure 12`**, TTT3R significantly flattens the error accumulation curve, reducing the drift rate by **\~2x**. It outperforms all standalone non-TTT baselines. Furthermore, TTT3R \+ Reset yields best performance.

| Method | Mechanism | Camera Pose (ATE$\\downarrow$) | Video Depth ($\\delta \< 1.25\\uparrow$)  |
| :---- | :---- | :---- | :---- |
| CUT3R | Standard Recurrent | 0.173 | 80.2 |
| \+ EMA | Exponential Moving Average | 0.164 | 80.7 |
| \+ BurnIn | Keyframe Updates | 0.144 | 80.2 |
| \+ Reset | Periodic Reset | 0.126 | 84.9 |
| **TTT3R** | **Confidence-guided Update** | **0.106** | **86.9** |
| **TTT3R \+ Reset** | **Combined** | **0.093** | **88.5** |

---

&nbsp;

**3\. Competitiveness with SOTA (R4)**
We compare TTT3R with the very recent **STream3R** (based on VGGT). While STream3R uses full-history attention (KV caching), TTT3R uses a compressed recurrent state.

**Result:** TTT3R achieves comparable accuracy to STream3R but with **significantly better efficiency**. Crucially, TTT3R maintains **constant memory usage (O(1))**, whereas STream3R suffers from linear memory growth (O(N)), making TTT3R uniquely suitable for indefinite streaming.

| Method | Memory Usage | Sintel Pose (ATE$\\downarrow$) | Sintel Pose (RPE rot$\\downarrow$) | BONN Depth (Abs Rel$\\downarrow$) | BONN Depth ($\\delta \< 1.25\\uparrow$) |
| :---- | :---- | :---- | :---- | :---- | :---- |
| STream3R | Growing $O(N)$ | 0.213 | 0.868 | 0.075 | 94.1 |
| **TTT3R** | **Constant $O(1)$** | **0.201** | **0.617** | **0.068** | **95.4** |

For more details, we provide point-by-point responses in each reviewer’s thread.

---

### Author Response · Authors · 2025-11-30
**General Response (1/2)**

We thank **R1(yvT7)**, **R2(jkPo)**, **R3(dBBg)**, **R4(6nKa)** for their constructive feedback. We are encouraged that reviewers recognize TTT3R as a **"novel"** (R1), **"elegant"** (R1, R2, R4), and **"highly significant"** (R1) contribution. Common recognition of our strengths includes:

1. *Significant Performance:*
   Achieving **"impressive"** (R2) and **"clearly better"** (R4) results, specifically a **"2x improvement"** in long-sequence pose estimation (R1, R3).
2. *Exceptional Efficiency:*
   A **"training-free, plug-and-play"** (R3, R4) design that maintains **"O(1) memory"** and **"real-time"** speed (R1, R3) in long sequences, featuring a lightweight nature that "**requires no additional parameters**" (R4).
3. *Insightful Framework:*
   A **"neat"** (R2) and **"clean"** (R3) reformulation of state updates as test-time training, offering a **"fresh"** (R4) perspective on catastrophic forgetting.

We address each concern with extensive additional experiments, including **6 new analysis subsections in the Appendix**, and revise the manuscript accordingly. Furthermore, TTT3R code will be released for future research.

All changes in the updated `PDF` are highlighted in **red**.

---

&nbsp;

## Part I: Summary of Revisions

**1\. Experimental Validation:**

* **Ablation on Learnable Gating:** Comparison with ScalarLR, ConditionLR, and TokenLR (R1Q2, R2Q3, R3Q1; `Tab.1`, `Fig.11`).
* **Non-TTT Baselines:** Comparison with Reset, EMA, and Burn-In mechanisms (R3Q4; `Tab.2`, `Tab.3`, `Tab.4`, `Tab.5`, `Fig.12`).
* **SOTA Comparisons:** Comparison with recent approaches like STream3R (R4Q3; `Tab.6`, `Tab.7`).
* **Method Combinations:** Evaluation of TTT3R \+ Reset (R3Q4, R4Q2; `Tab.5`, `Fig.12`, `Fig.13`, `Fig.14`, `Fig.15`).
* **Finetuning Analysis:** Results of finetuning TTT3R (R2Q2; `Tab.1`, `Fig.11`).

**2\. Methodological Clarifications:**

* **Explicit Formulation:** We add the final update equation (R1Q1; `L.322`, `L.328`).
* **Intuitive Explanation:** We clarify "per-token adaptive weighting" vs. TTT gradients (R2Q1; `L.267`, `L.296`, `L.305`, `L.312`).
* **Gradient & Causality:** We clarify the state update as a meta-learned, strictly causal implicit gradient step implemented via cross-attention (R3Q2; `L.296`, `L.322`).
* **Terminology:** We refine the language from "analyze dynamics" to "analyze behavior" (R3Q3; `L.104`).

**3\. Extended Discussions & Related Work:**

* **Generality of Methodology:** We discuss applicability to other reconstruction foundation models beyond CUT3R (R4Q1; `L.223`, `L.229`).
* **Comparison with Standard TTT:** We discuss conceptual differences (R1Q2, R3Q1; `L.901`, `L.915`).
* **Gap with Global-Attention:** We analyze efficiency-accuracy trade-offs (R4Q2; `L.527`, `L.1063`).
* **New Related Works:** We discuss Test3R, MegaSaM, VIPE, VGGT-SLAM, and VGGT-Long (R1Q3, R4Q3; `L.117`, `L.184`).

---

### Meta-Review · Area_Chair_waRc · 2026-01-02

**Summary:**

This paper proposes an effective training-free approach, TTT3R, to improves length generalization. All reviewers appreciate the well-motivated idea, training-free property, and practicality of the proposed method. However, they also raise lots of concerns including the limited novelty and poor performance against the offline methods.

**Reviewer Concerns:**

The major differences from the CUT3R should be clarified. The discussions with recent approaches, such as MegaSaM, VIPE, VGGT-SLAM, and VGGT-Long should be provided.

**Reviewer Scores:**

Based on the provided rebuttals, the authors clarified the major differences from CUT3R. In addition, the authors provide comparisons with STream3R.

---

### Decision · Program_Chairs · 2026-01-26

Accept (Poster)